# Lightweight deep learning for real-time road distress detection on mobile devices

Yuanyuan Hu [1,6], Ning Chen[2,6], Yue Hou [3] ✉, Xingshi Lin[4], Baohong Jing[5] & Pengfei Liu [1] ✉

Efficient and accurate road distress detection is crucial for infrastructure maintenance and transportation safety. Traditional manual inspections are labor-intensive and time-consuming, while increasingly popular automated systems often rely on computationally intensive devices, limiting widespread adoption. To address these challenges, this study introduces MobiLiteNet, a lightweight deep learning approach designed for mobile deployment on smartphones and mixed reality systems. Utilizing a diverse dataset collected from Europe and Asia, MobiLiteNet incorporates Efficient Channel Attention to boost model performance, followed by structural refinement, sparse knowledge distillation, structured pruning, and quantization to significantly increase the computational efficiency while preserving high detection accuracy. To validate its effectiveness, MobiLiteNet improves the existing MobileNet model. Test results show that the improved MobileNet outperforms baseline models on mobile devices. With significantly reduced computational costs, this approach enables real-time, scalable, and accurate road distress detection, contributing to more efficient road infrastructure management and intelligent transportation systems.

Road infrastructures play a fundamental role in modern public transportation[1,2]. Throughout the service life, they endure repeated vehicle loads and severe environmental influences, resulting in distress such as cracks, rutting, and potholes[3,4]. If the problems are not addressed, the existing road distresses can further develop into structural failures, leading to significantly higher repair costs[5,6]. Thus, for road authorities all over the world, it is crucial to have regular road inspections and timely maintenance. However, traditional methods, which rely heavily on manual surveys and specialized equipment, remain labor-intensive, slow, and prone to errors[7]. These problems also limit the scalability and fail to meet the demands of very large road networks. Consequently, there is an urgent need for automated, efficient, and real-time road monitoring systems that can provide accurate and scalable solutions.

In recent years, various automated road distress detection methods have emerged and been developed, including traditional image processing techniques such as thresholding[8,9] and edge detection[10,11]. While these methods provided a preliminary approach to improve the productivity of road distress detection, the effectiveness is often limited by the negative factors during inspection, including variations in camera lighting, very complex road material texture, significant background noise on the road surface, etc. The development of machine learning[12–16], especially deep learning[17,18], has significantly improved the current road distress detection methods[19–26]. These advanced detection methods require substantial computational resources, leading to different implementation strategies for practical deployment. Generally, current automated detection approaches can be categorized into cloud-based and mobile edge-computing systems.

[1]Institute of Highway Engineering, RWTH Aachen University, Aachen, Germany. [2]Beijing Key Laboratory of Traffic Engineering, Beijing University of Technology, Beijing, China. [3]Department of Civil Engineering, Faculty of Science and Engineering, Swansea University, Swansea, UK. [4]Fujian Yongzheng Construction Quality Inspection Co., Ltd., Fuzhou, China. [5]Qingdao Yicheng Sichuang Link of Things Technology Co., Ltd., Qingdao, China. [6]These authors contributed equally: Yuanyuan Hu, Ning Chen. ✉e-mail: yue.hou@swansea.ac.uk; liu@isac.rwth-aachen.de

In cloud-based systems, high-resolution road inspection data, which amounts to tens of terabytes per session due to the extensive mileage covered, must be uploaded to the cloud for processing. This creates substantial challenges in terms of data transmission and storage, and raises concerns about data security, privacy, and the limited real-time responsiveness of such systems[27,28]. Edge computing, in contrast, refers to the practice of processing data at or near the source of data generation rather than relying on centralized cloud servers[29]. By enabling real-time, on-site data processing directly on mobile devices, edge computing reduces dependency on cloud infrastructures[30,31]. Edge computing addresses privacy concerns, minimizes transmission latency, and significantly lowers the bandwidth requirements by storing or uploading only the detection results[32]. Consequently, it drastically alleviates storage burdens while ensuring low-latency performance, making it particularly suitable for high-speed and large-scale road distress inspections.

Mobile edge computing devices, primarily including smartphones and mixed reality (MR) devices, offer unique advantages for real-time road distress detection[33]. Smartphones, equipped with advanced processors, cameras, and connectivity features, serve as powerful tools for on-site data collection and processing. Their portability, user-friendly interfaces, and widespread adoption reduce hardware costs and enable scalable deployment in mobile edge-computing systems[34]. MR devices, which integrate augmented and virtual reality, are particularly valuable for specialized applications such as on-site maintenance planning, immersive infrastructure assessments, and augmented reality-assisted crack sealing, where visual overlays enhance decision-making for engineers and maintenance personnel[35,36]. As a supplement to existing detection tools, mobile devices enhance flexibility and versatility, making them applicable in scenarios where traditional methods may not be suitable. For example, they allow inspectors to conduct assessments in confined or hard-to-reach areas, perform emergency inspections after natural disasters, and operate in hazardous environments where deploying large equipment is impractical[37,38].

Despite its significant potential, mobile edge computing technology remains underutilized in road infrastructure monitoring and management. A key challenge is the absence of a specially designed framework that supports the deployment of lightweight models optimized for real-time road distress detection. Current approaches often struggle to achieve both efficiency and accuracy on devices with restricted computational power and battery life, limiting their practical application in mobile environments. Another major limitation is the lack of high-quality datasets specifically designed for mobile environments, hindering the development and implementation of effective solutions. Addressing these gaps is crucial for realizing the full potential of mobile edge computing technology in road infrastructure management.

To address all these issues, in this work, we present MobiLiteNet, a lightweight deep learning framework designed for real-time road monitoring on mobile devices. The proposed approach integrates advanced model optimization techniques, including efficient channel attention (ECA) mechanisms[39], structural refinement, sparse knowledge distillation[40], structured pruning[41], and quantization[42], enabling high detection accuracy with significantly reduced computational demands. This design facilitates deployment on resource-constrained devices such as smartphones and MR systems, enhancing the practicality of automated road distress detection in real-world scenarios. The MobiLiteNet framework, when integrated with MobileNet V2 architecture—a well-established model for efficient computer vision tasks on edge devices—is built upon a diverse dataset collected across European and Asian regions to enhance robust performance across varied road conditions. This approach achieves enhanced performance compared to the original model, delivering both higher accuracy and lower inference latency across different deployment platforms. The framework not only advances the current state of mobile-based infrastructure monitoring but also lays the groundwork for next-generation automated assessment

systems, offering broad potential applications in intelligent transportation, smart city initiatives, and inclusive technologies designed to enhance public safety and accessibility.

The main contributions of this study are as follows:
1. The MobiLiteNet framework is proposed, integrating advanced deep learning optimization techniques to achieve efficient and accurate road distress detection suitable for smartphones and MR devices.
2. A diverse dataset is constructed, consisting of road distress images collected from representative regions in Europe and Asia, capturing a wide range of real service conditions, thereby enhancing the robustness and generalization capability of the trained models.
3. The optimized MobileNet V2 model, developed through the MobiLiteNet framework and trained on a diverse dataset, demonstrates effective performance in field deployment on smartphones and MR devices in Aachen, Germany, validating its computational efficiency and detection accuracy for real-time road monitoring in complex environments.

## Results
### Workflow of experimental and results overview
To evaluate the effectiveness of the proposed MobiLiteNet for real-time road monitoring, comprehensive experiments were conducted covering model development, deployment, and real-world validation. The overall experimental workflow is illustrated in Fig. 1, which shows the entire process from data collection and augmentation to on-site validation.

The research began with data collection and augmentation, where road images were gathered from diverse geographic locations in Europe and Asia under varying environmental conditions. To enhance dataset robustness, data preprocessing techniques and advanced augmentation strategies were employed to simulate diverse crack patterns under different weather conditions. Following data augmentation, the MobiLiteNet framework was developed with a two-stage approach. First, ECA[39] mechanisms are employed to enhance model performance. Subsequently, structural refinement, sparse knowledge distillation[40], structured pruning[41], and quantization[42] are applied to reduce computational complexity while preserving high detection accuracy. The optimized model was subsequently converted into TensorFlow Lite (TFLite) format for deployment on smartphones and MR devices. To validate the system, it was field-tested in real-world engineering projects in Aachen, Germany, to evaluate its generalization capabilities. Collected data is planned to be later uploaded to a cloud server and systematically integrated into the future public Mobilithek database, a centralized platform initiated by the German Federal Ministry for Digital and Transport (BMDV) to standardize and streamline national mobility data. The integration of mobile edge computing technology enhances data collection precision and supports efficient road management, contributing to continuous improvement and scalability of the monitoring system.

The subsequent sections present detailed experimental results, beginning with an in-depth analysis of the MobiLiteNet's architecture and optimization techniques. The performance of the proposed model is compared against the baseline model (original MobileNet V2), with a focus on three key metrics: detection accuracy, model architecture, and parameter size. This comparison aims to demonstrate how the architectural improvements and optimizations within MobiLiteNet contribute to improved efficiency and accuracy. An ablation study was then conducted to analyze the contributions of each component to the overall performance, identifying the specific impact of individual techniques incorporated in MobiLiteNet. Following this systematic evaluation, deployment results on smartphones and MR devices are discussed to illustrate the framework's generalization capabilities across different environments. Finally, the effectiveness of the system

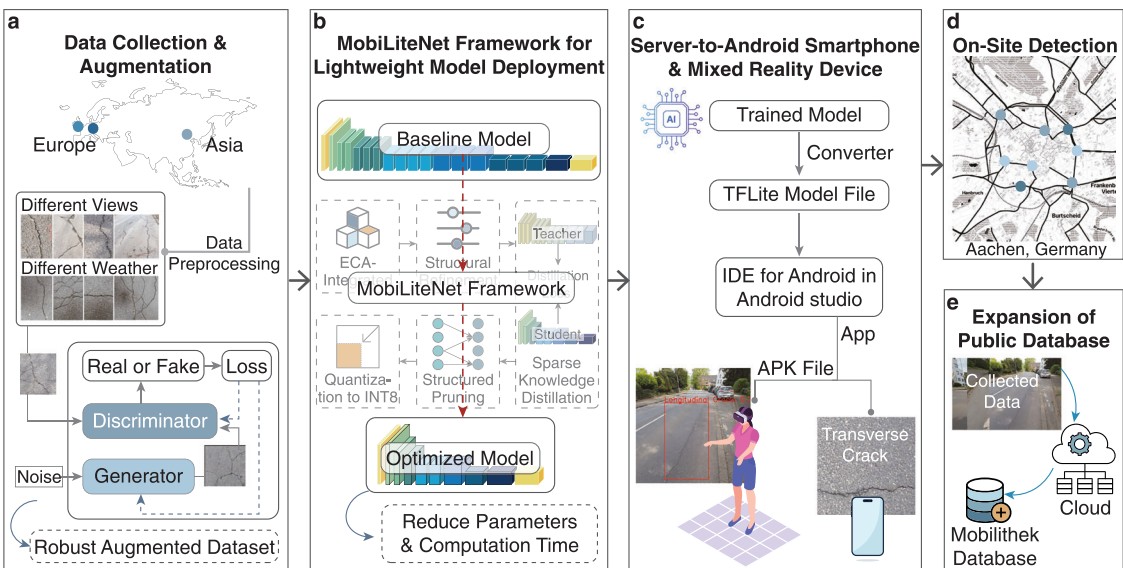

**Fig. 1 | Experimental workflow for mobile-based road monitoring.** Overview of data processing, model optimization, mobile deployment, and on-site detection. **a** data collection and augmentation. Icon by Leonardo Henrique Martini from The Noun Project (CC BY 3.0). **b** MobiLiteNet Framework for Lightweight Model Deployment. Icon by HideMaru from The Noun Project (CC BY 3.0). **c** Server-to- android smartphone and mixed reality device. Contains elements by juicy_fish and macrovector via Freepik Free License. **d** On-site detection. Map tiles adapted from Stamen Design under CC BY 3.0. Data © OpenStreetMap contributors (ODbL). **e** Expansion of public database. Icons by Tini Sumiarsih and Alzam from The Noun Project (CC BY 3.0).

was validated through on-site experiments, showcasing its practical applicability in real-world road monitoring scenarios.

## The proposed MobiLiteNet framework

As illustrated in Fig. 2a, MobiLiteNet is designed through a series of systematic optimizations that first maximize the model's representational capacity and detection accuracy, and then systematically reduce computational resources for real-time mobile deployment. This sequential approach ensures robust performance in the initial phase, followed by lightweight optimizations that facilitate efficient on-site operation. To demonstrate these principles in practice, Fig. 2b provides a detailed example of how MobileNet V2 is optimized and refined within the MobiLiteNet framework. Specifically, it integrates five critical components—ECA mechanisms[39], structural refinement, sparse knowledge distillation[40], structured pruning[41], and quantization[42]—thereby obtaining a balance between high performance and efficient resource utilization on smartphones and MR devices.

The process begins with the integration of ECA mechanisms, a lightweight module designed to enhance the model's ability to capture essential channel-wise dependencies with minimal computational resources. Unlike traditional attention mechanisms, ECA employs simple one-dimensional convolution operations, efficiently modeling cross-channel interactions[39].

Following this, structural refinement is employed to optimize the model architecture. By reducing the number of bottleneck repetitions and adjusting the input–output channel dimensions, this step significantly decreases the number of parameters and computational load, thus improving the model's efficiency while maintaining its detection capability.

To further improve model efficiency, sparse knowledge distillation[40] is introduced, where a high-capacity ResNet[43] teacher model transfers its learned knowledge to a lightweight student model. Knowledge distillation[40] provides the advantage of enabling the student model to inherit the generalization capabilities of the teacher model while significantly reducing model complexity. This transfer process is facilitated through a combination of soft target loss, which captures the probabilistic outputs of the teacher model to convey nuanced inter-class relationships, and hard target loss, which ensures

the model maintains strong performance on ground-truth labels. Additionally, an L1 norm regularization[44] term is incorporated into the distillation loss to promote sparsity in the model's parameters. This sparsity-inducing regularization not only enhances model compression but also lays a solid foundation for subsequent structured pruning operations by making it easier to identify and eliminate less critical parameters. This approach allows the student model to achieve performance levels comparable to the teacher model while maintaining a compact structure highly suitable for mobile deployment.

Subsequently, structured pruning[41] is applied to eliminate less significant channels based on their contribution to model performance. By systematically removing up to 30% of the channels, this step significantly reduces the number of parameters, leading to a leaner model architecture. The reduction in parameter count not only decreases memory usage but also lowers computational demands, which in turn accelerates inference times.

Following pruning, the model undergoes quantization[42], converting floating-point operations (float32) to 8-bit integer (INT8) representations. This conversion reduces the model size to approximately one-fourth of its original size, significantly decreasing memory usage. Quantization not only reduces the storage requirements but also enhances processing efficiency by enabling faster arithmetic operations that are optimized for mobile hardware. This substantial reduction in computational complexity is crucial for achieving real-time distress detection capabilities, as it minimizes latency and ensures smooth performance in dynamic and on-site environments.

In sum, these optimization techniques within MobiLiteNet, as applied to the MobileNet V2 and shown in Fig. 2, enable the deployment of deep learning models on smartphones and MR devices, supporting real-time and on-site road distress detection with high accuracy and efficiency.

## Dataset construction and augmentation

Existing datasets such as GAPs[45–47], CRACK500[48], and CrackForest[49] have made significant contributions to road distress detection by providing high-quality annotated images. However, these datasets primarily focus on images captured under clear or overcast conditions, limiting their applicability across diverse environmental scenarios. This constraint

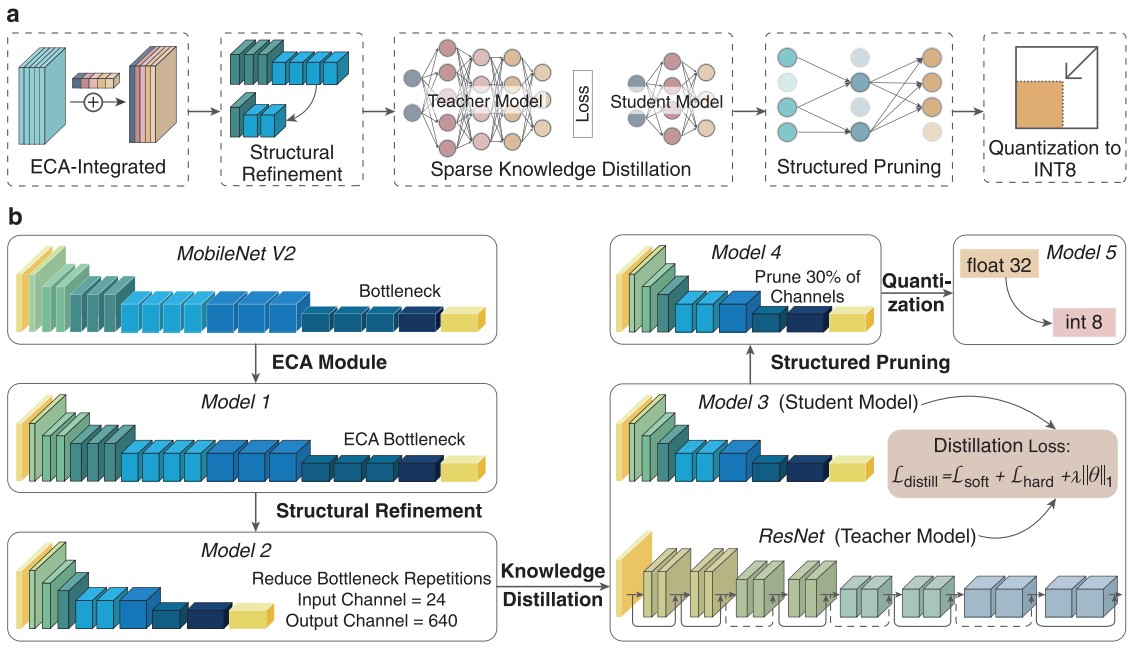

**Fig. 2 | Architecture optimization in the MobiLiteNet framework. a** Process schematic for the MobiLiteNet architecture. **b** Architecture and implementation details based on MobileNet V2. ECA efficient channel attention.

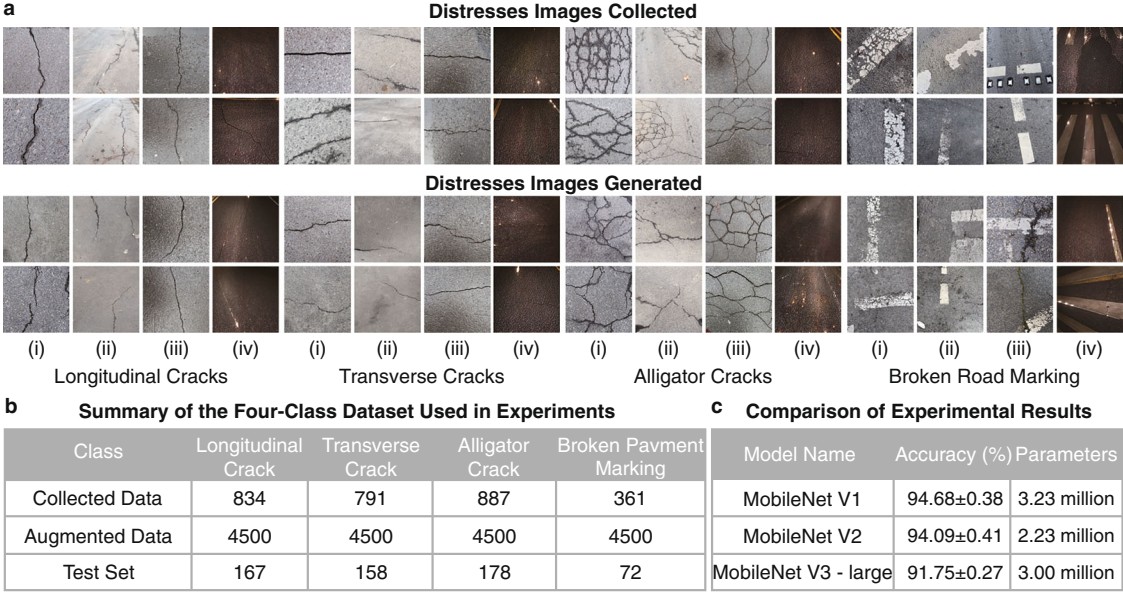

| Class | Longitudinal Crack | Transverse Crack | Alligator Crack | Broken Pavement Marking |
|---|---|---|---|---|
| Collected Data | 834 | 791 | 887 | 361 |
| Augmented Data | 4500 | 4500 | 4500 | 4500 |
| Test Set | 167 | 158 | 178 | 72 |

| Model Name | Accuracy (%) | Parameters |
|---|---|---|
| MobileNet V1 | 94.68±0.38 | 3.23 million |
| MobileNet V2 | 94.09±0.41 | 2.23 million |
| MobileNet V3 - large | 91.75±0.27 | 3.00 million |

**Fig. 3 | Summary of dataset composition and augmentation. a** Collection and generation of images of road distress. Categorized into four types: longitudinal cracks, transverse cracks, alligator cracks, and broken pavement markings. Each category includes images captured under four conditions: (i) parallel view, (ii) oblique view, (iii) rainy weather, and (iv) nighttime. **b** Dataset used in experiments. **c** Comparison of experimental results for the MobileNet family models.

presents a challenge for developing robust detection systems capable of functioning reliably in real-world deployment environments where lighting and weather conditions vary considerably.

To address these limitations and build a robust, diverse dataset for training and validating the proposed MobiLiteNet framework, road distress images were collected from multiple geographic locations, including Aachen in Germany, Beijing in China, and Swansea in the United Kingdom. While the general classification of road distress remains similar across these regions, their specific characteristics vary due to differences in temperature variations, moisture exposure, and traffic loads. Aachen experiences moderate seasonal temperature variations and high traffic intensity, particularly on arterial roads and

highways, leading to distress patterns influenced by both thermal fluctuations and heavy dynamic loads. Beijing, characterized by extreme seasonal temperature shifts, undergoes significant thermal expansion and contraction cycles, impacting pavement durability, especially in resurfaced layers. Swansea, with its high humidity and frequent rainfall, faces persistent moisture exposure, making water-related deterioration mechanisms more prevalent. This diversity significantly enhances the dataset's generalization capability for real-world deployment. To account for a wider range of environmental conditions, the dataset was expanded to include images taken during rainy weather and at night, as shown in Fig. 3a, ensuring the model's robustness across different environmental conditions. Additionally,

images were captured from various viewing angles, including both parallel and oblique perspectives relative to the pavement, to simulate real-world data acquisition scenarios and improve model performance under diverse viewpoints. The dataset covers four classes of road distresses: longitudinal cracks, transverse cracks, alligator cracks, and broken road markings. A total of 2873 road distress images were collected and processed to a standardized resolution of $512 \times 512$ pixels, ensuring consistency for model training.

To have a larger dataset size that is more convenient for machine learning calculations, a Wasserstein Generative Adversarial Network with Gradient Penalty (WGAN-GP)[50]. was employed. As shown in Fig. 3a, WGAN-GP was capable of generating high-quality synthetic images for each distress category, enhancing dataset diversity and addressing data imbalance issues. The collected data was initially divided in a 4:1 ratio, with the larger portion used for augmentation and the remainder reserved for testing. For each distress class, 4500 synthetic images were generated, of which 4000 were allocated for training and 500 for validation, as summarized in Fig. 3b. This data augmentation process further improves model robustness and generalization by addressing data scarcity and enhancing adaptability to diverse road conditions.

## Model optimization and performance evaluation

The MobileNet series[51–53], a family of lightweight convolutional neural networks (CNNs) designed for efficient deep learning, is renowned for its computational efficiency, making it particularly well-suited for deployment on embedded systems and mobile devices where computational resources are limited[54,55]. Among its variants, MobileNet V2 obtains the optimal balance between accuracy and resource efficiency, as shown in Fig. 3c. Its core structure relies on an inverted residual bottleneck module, which consists of depthwise separable convolutions that significantly reduce computational cost while maintaining performance[52]. This design expands the input channels before applying depthwise convolutions and then projects them back to a lower-dimensional space, enhancing feature representation efficiency. Despite its efficiency, MobileNet V2 presents potential for further optimization, particularly for the complex challenges of road distress detection in resource-constrained environments. This opportunity for enhancement makes it an ideal baseline model for developing and benchmarking the proposed MobiLiteNet framework.

The architecture of the optimized MobiLiteNet-based improved MobileNet V2 model is illustrated in Fig. 4, where Fig. 4a represents the original MobileNet V2 architecture, and Fig. 4b shows the optimized model after applying the MobiLiteNet framework. This optimization led to a significant reduction in the number of model parameters, decreasing from 2,228,996 to 498,283 parameters. Remarkably, this reduction in model complexity did not compromise performance; instead, the model's accuracy improved from 94.09% to 96.38%.

The performance improvement of the MobiLiteNet framework can be attributed to the synergistic effects of ECA mechanisms, structural refinement, sparse knowledge distillation, structured pruning, and quantization. Among these methods, sparse knowledge distillation plays a key role by transferring high-level knowledge from a highly accurate teacher model to a lightweight student model, enhancing generalization while introducing sparsity through L1 norm regularization. This sparsity facilitates the subsequent structured pruning process, which systematically removes redundant parameters to reduce computational complexity without compromising model accuracy. The ECA mechanism further contributes by enhancing feature representation through efficient channel-wise attention, allowing the model to focus on critical features with minimal computational resources. Furthermore, quantization converts model weights from float32 to INT8, significantly reducing memory usage and accelerating inference speed by compressing the model size to one-fourth of its original. Overall, the results demonstrate the effectiveness of the MobiLiteNet framework in enhancing both efficiency and accuracy.

## MobiLiteNet deployment workflow and component-wise ablation analysis

To evaluate the practical applicability of MobiLiteNet, comprehensive deployment tests and ablation studies were conducted to analyze the contributions of individual optimization components to the framework's overall performance.

The deployment process is illustrated in Fig. 5a, detailing the complete workflow from model optimization to real-world application. Following model training, the optimized MobileNet V2 model based on MobiLiteNet was converted into TFLite format[56,57], which is specifically designed for efficient execution on resource-constrained devices. This conversion is a critical step for mobile deployment, as it optimizes the model architecture for inference on edge devices while preserving detection capabilities. The development environment was configured using Android Studio, where the TFLite model was integrated into the custom-developed Android application named Road-Intelligent. This application serves as an intuitive interface for real-time road distress detection, capturing images through both smartphone and MR device cameras. Each captured image undergoes a preprocessing pipeline that includes resizing to the model's input dimensions ($512 \times 512$ pixels), normalization to standardize pixel values, and format conversion for model compatibility. The preprocessed data is then fed into the model for inference, with detection results immediately displayed on the application interface. This immediate feedback mechanism enables users to promptly identify various road distresses during field inspections.

To systematically evaluate the contributions of each optimization component in MobiLiteNet, a series of ablation studies was conducted. These experiments analyze the impact of individual components—ECA modules, structural refinement, knowledge distillation, structured pruning, and quantization—on the model's performance. The detailed results are presented in Fig. 5b, c.

Figure 5b summarizes the model configurations and performance metrics, highlighting which optimization techniques are incorporated in each variant. ResNet demonstrates superior accuracy ($98.02 \pm 0.20\%$) due to its complex architecture. However, it suffers from high computational costs with over 11 million parameters and a model size of 44.7 MB, rendering it impractical for mobile deployment. Nevertheless, its strong performance substantiates the selection of ResNet as the teacher model in the knowledge distillation process. In contrast, the baseline MobileNet V2 showed significantly reduced computational demands (2.229 million parameters, 8.9 MB) but achieved lower accuracy ($94.09 \pm 0.41\%$) compared to ResNet. This performance gap indicated the need for architectural enhancements while maintaining MobileNet V2's computational efficiency advantages for mobile deployment.

The subsequent models were derived by incrementally incorporating components of the MobiLiteNet framework. By integrating the ECA module into MobileNet V2 (Model 1), feature representation capabilities were enhanced, resulting in improved accuracy ($96.10 \pm 0.34\%$) without increasing the parameter count. This demonstrates the effectiveness of attention mechanisms in strengthening feature extraction with minimal computational resources. The addition of structural refinement (Model 2) reduced the number of parameters by approximately 55% (from 2.229 to 0.996 million) and decreased model size to 4.0 MB. This significant reduction in model complexity resulted in a temporary accuracy decrease to $93.43 \pm 0.38\%$. Incorporating sparse knowledge distillation with ResNet as the teacher model (Model 3) substantially improved generalization capabilities, achieving an accuracy of $97.15 \pm 0.29\%$. This highlights the critical role of knowledge transfer in enhancing the model's learning capability while maintaining the reduced parameter

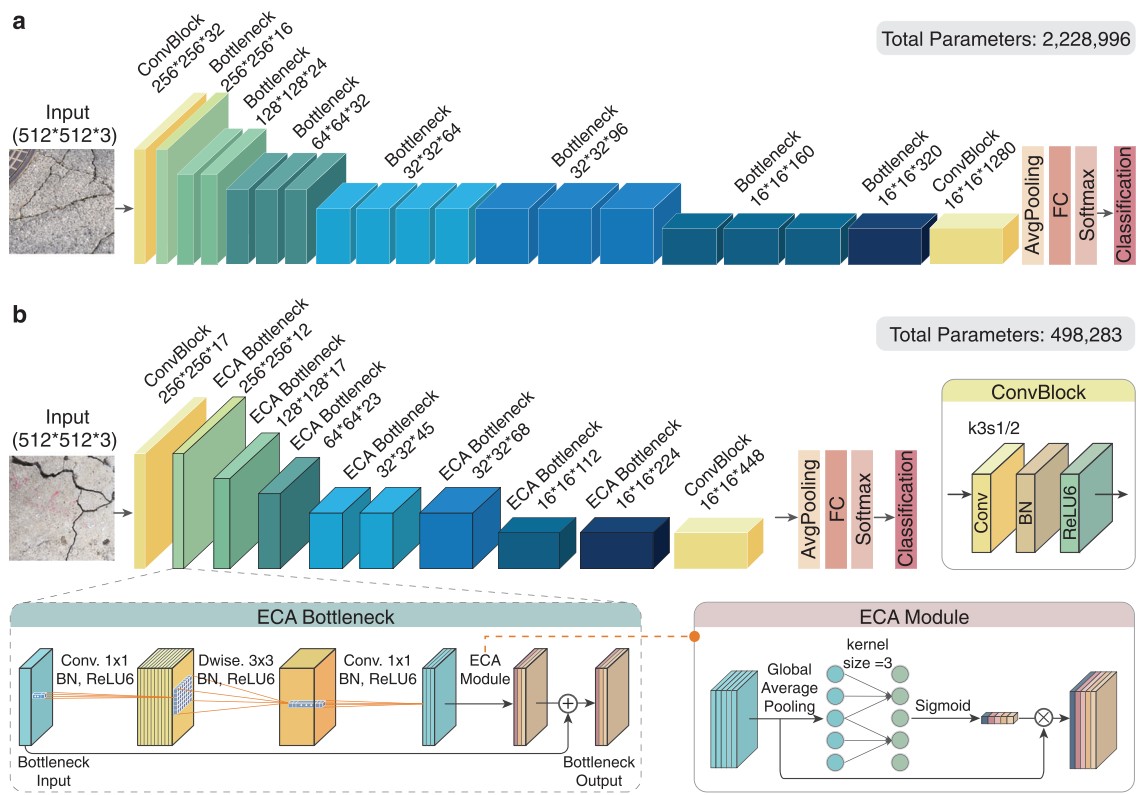

**Fig. 4 | Structural optimization of the baseline model using the MobiLiteNet framework. a** Original structure of MobileNet V2. **b** Optimized model with MobiLiteNet framework.

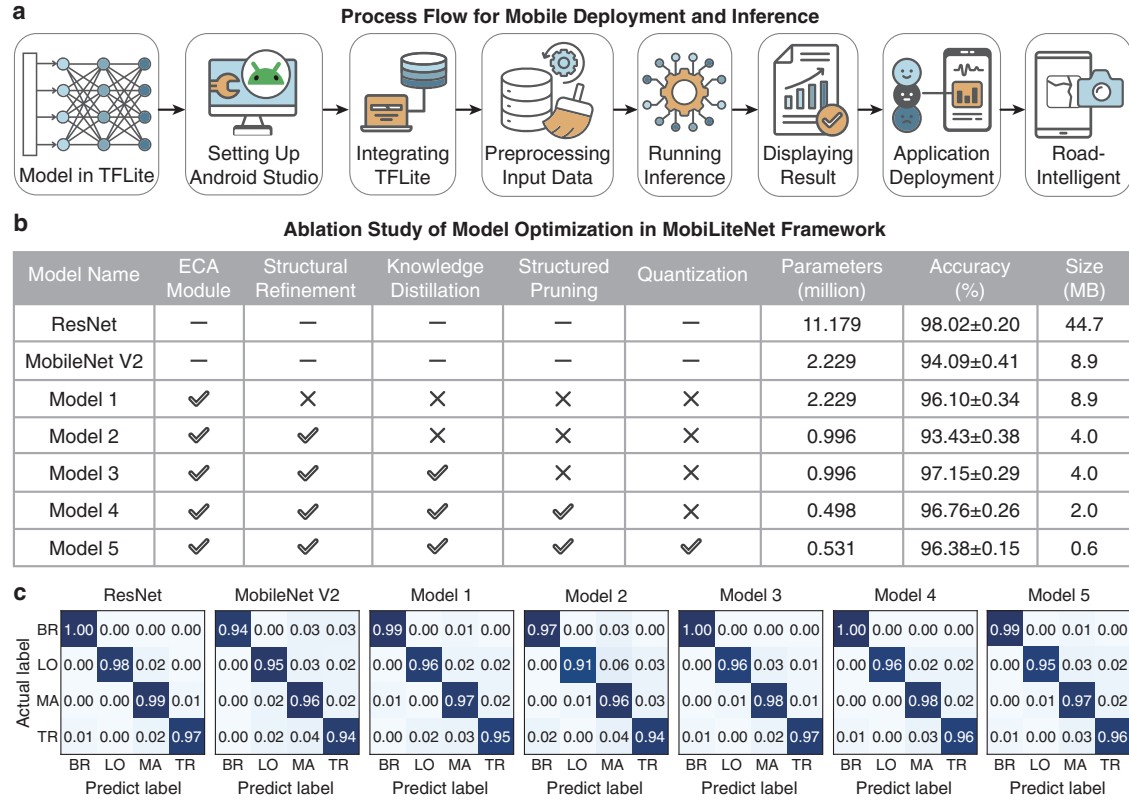

**Fig. 5 | End-to-end deployment and ablation study of MobiLiteNet framework. a** Process flow for mobile deployment and inference. Icons by Lucas Rathgeb, Puspito, Camallia Marroh, Good Wife, Dan's, Ali Mahmudi, Ainul Abib, and Reza Nur from the Noun Project (CC BY 3.0). **b** Ablation study of model optimization in MobiLiteNet framework. **c** Confusion matrices for ablation study models: BR broken pavement marking, LO longitudinal crack, MA alligator crack, TR transverse crack.

**a**  **Smartphones and Mixed Reality Devices**

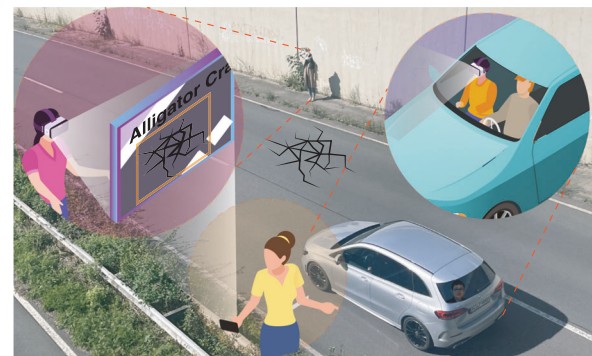

**b**  **End-to-End Processing Time Comparison of Different Models on Mobile Devices**

|  | ResNet | MobileNet V2 | Model 1 | Model 2 | Model 3 | Model 4 | Model 5 |
|---|---|---|---|---|---|---|---|
| Speed of Smartphones (ms) | 592.6 | 122.3 | 122.5 | 71.5 | 70.7 | 47.2 | 34.5 |
| Speed of MR Devices (ms) | 1450.4 | 288.9 | 289.9 | 129.6 | 128.5 | 64.8 | 40.1 |

**Fig. 6 | Practical deployment and performance comparison on mobile devices. a** Smartphone-based and MR-based application deployment visualization. Contains elements by macrovector via Freepik Free License. **b** End-to-end processing time comparison of different models on mobile devices.

count. The application of structured pruning (Model 4) further reduced the parameter count to 0.498 million and model size to 2.0 MB, with minimal impact on accuracy (96.76 ± 0.26%). This demonstrates the effectiveness of systematic parameter reduction in maintaining detection capabilities while improving computational efficiency. The final model (Model 5) incorporates quantization by converting weights to INT8 format, reducing the model size to merely 0.6 MB while maintaining robust detection capabilities (96.38 ± 0.15%). Compared to the baseline MobileNet V2, Model 5 shows a statistically significant improvement in accuracy while achieving a model size reduction to approximately one-fifteenth of the original.

Figure 5c presents confusion matrices (typical) for each model, illustrating their classification performance across different road distress categories. The results demonstrate that each optimization step contributes meaningfully to maintaining or enhancing detection accuracy while improving computational efficiency. Notably, Model 5 achieves performance comparable to ResNet while significantly reducing computational requirements, highlighting the MobiLiteNet framework's efficiency for real-time, resource-constrained applications. The ablation studies demonstrate that each component of the framework contributes critically to optimizing model performance, ensuring a balanced trade-off between accuracy, speed, and model size for effective mobile deployment.

### Cross-platform deployment and performance analysis on smartphones and MR devices

The optimized MobileNet V2 model based on MobiLiteNet was successfully deployed on both smartphones and MR devices through Android application packages (APKs) generated using Android Studio, as illustrated in Fig. 6a.

For smartphone deployment, the RoadIntelligent application was developed to enable users to capture images, process them through the optimized model, and display detection results in real-time. This implementation utilizes the smartphone's built-in camera for image acquisition, with the detection results visualized directly on the screen, providing immediate feedback on road conditions.

In parallel, the optimized model was also deployed on MR devices through a specialized implementation of the RoadIntelligent application adapted for the MR environment. The MR version of RoadIntelligent enables direct visualization of detection results as overlays on the physical road surface. This implementation features gesture-

based interface controls for hands-free operation, allowing inspectors to examine, categorize, and document distress without interrupting their workflow. The spatial registration capabilities of the RoadIntelligent MR application provide immediate contextual information while preserving the computational efficiency of the underlying MobiLiteNet-optimized model.

To evaluate the performance across different hardware platforms, end-to-end processing time was measured for each model variant on both smartphones and MR devices, as presented in Fig. 6b. For these speed tests, ten randomly selected images from test set consistent across all evaluations were used, with the reported values representing the average processing times. The processing time includes image preprocessing, model inference, and result visualization. On smartphones, the fully optimized Model 5 achieved an impressive processing time of 34.5 ms, representing a 71.8% reduction compared to the baseline MobileNet V2 (122.3 ms) and an overwhelming 94.2% reduction compared to ResNet (592.6 ms). Similar performance improvements were observed on MR devices, where Model 5 achieved a processing time of 40.1 ms, representing an 86.1% reduction compared to MobileNet V2 (288.9 ms) and a 97.2% reduction compared to ResNet (1450.4 ms). The results demonstrate that each progressive optimization step in the MobiLiteNet framework contributes to substantial improvements in processing efficiency.

The MobiLiteNet-optimized model delivers several key benefits for mobile-based road distress detection. The reduced processing time (34.5 ms on smartphones, 40.1 ms on MR devices) enables real-time detection at operational speeds critical for efficient field inspections. The smaller model size (from 8.9 MB to 0.6 MB) allows deployment across diverse hardware configurations while minimizing storage requirements. These optimizations extend device battery life during field operations and enable concurrent execution with other applications without performance degradation. The significant improvements in processing efficiency (71.8% reduction on smartphones, 86.1% on MR devices) directly translate to reduced operational costs, enhanced accessibility, and improved scalability for practical road monitoring implementations.

### Field validation using smartphones and MR devices

To evaluate the practical effectiveness of the model, extensive field validations were conducted in Aachen, Germany, as illustrated in Fig. 7a and demonstrated in Supplementary Movies 1 and 2. The tests

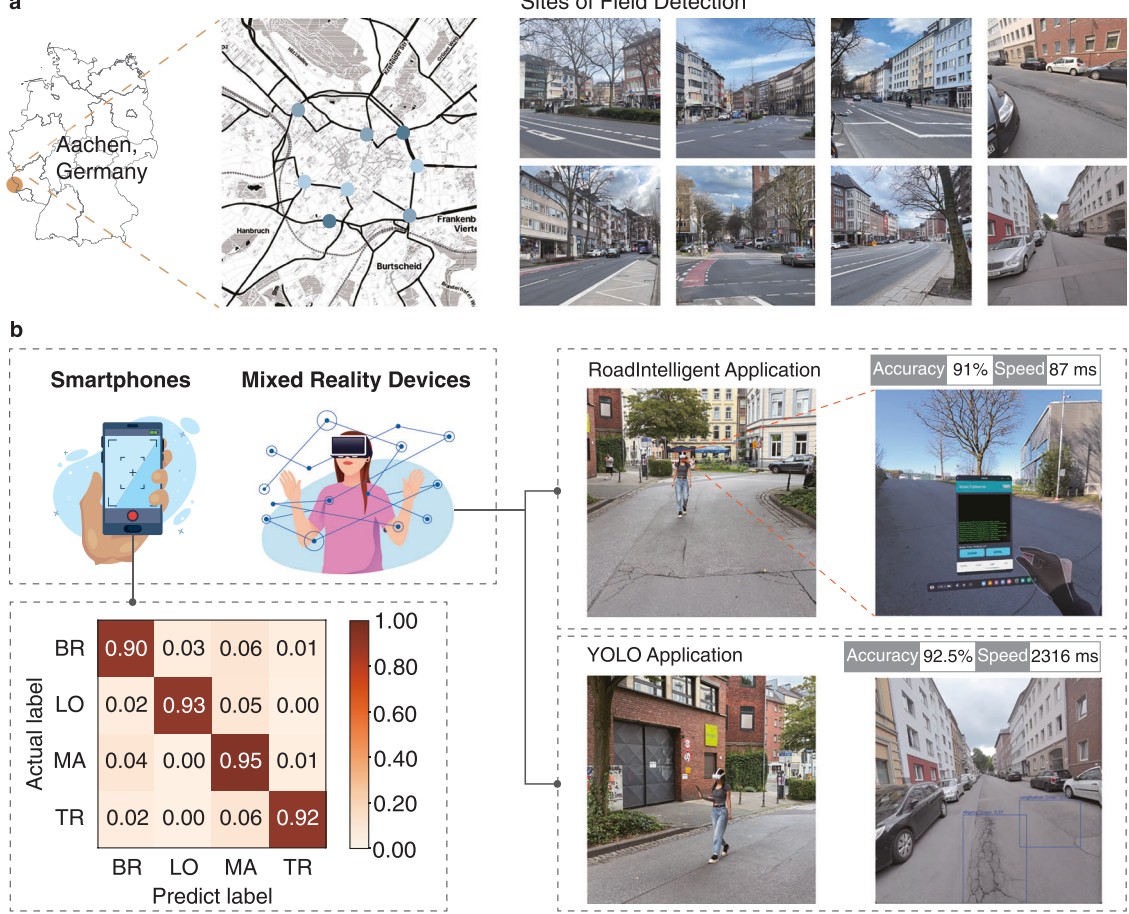

**Fig. 7 | Experimental setup and performance for real-world road monitoring.**
**a** Geographic distribution of field detection sites in Aachen. Map tiles adapted from
Stamen Design under CC BY 3.0. Data © OpenStreetMap contributors (ODbL). **b** Detection performance metrics across devices and modes. Contains elements by studiogstock and freepik via Freepik Free License.

were performed using the same smartphone and MR device configurations that were utilized during the training and development phases, ensuring consistency in the assessment of real-world performance.

For smartphone-based validation, a comprehensive dataset of 400 road distress images was collected across various locations in Aachen, equally distributed among the four distress categories: 100 images each of longitudinal cracks, transverse cracks, alligator cracks, and broken road markings. The detection results are summarized in the confusion matrix presented in Fig. 7b, which demonstrates high classification accuracy across all categories. The overall accuracy achieved was 92.5%. These results validate the model's robust performance in real-world conditions, despite variations in lighting, camera angles, and environmental factors that differ from the controlled training environment.

The MR-based validation was conducted using the same field sites but employed a different data collection approach suitable for the MR platform. Besides the RoadIntelligent application (see Supplementary Fig. 1), a complementary MR-based YOLO application using YOLOv8 (see Supplementary Fig. 2) was developed to provide a comprehensive evaluation of the proposed MobiLiteNet framework in real-world scenarios through comparative analysis. The dataset preparation for this YOLO implementation is detailed in the Supplementary Information document. This state-of-the-art object detection model serves as a benchmark to assess the RoadIntelligent application's balance between detection accuracy and computational efficiency in field conditions.

A total of 200 high-resolution images (1440 × 1440 pixels) were captured and analyzed, with 50 images for each of the four distress categories. Both the RoadIntelligent application, based on the optimized MobileNet V2 model, and the YOLO application, based on YOLOv8, were used for comparative analysis. It should be noted that these applications perform different types of tasks—RoadIntelligent performs classification while YOLO performs object detection—and were trained on different datasets. The RoadIntelligent dataset was entirely self-collected, while YOLO utilized both field-acquired and publicly available data, with transportation engineers standardizing annotation criteria across both systems. Thus, the comparison provides an engineering reference rather than a direct functional equivalence. The detection accuracy and processing speeds of both applications are presented in Fig. 7b. The RoadIntelligent application achieved 91% accuracy with significantly faster detection times (87 ms), while the YOLO application reached 92.5% accuracy but required substantially longer processing times (2316 ms). Despite the marginally lower accuracy (1.5% difference), the RoadIntelligent application processed images nearly 27 times faster than the YOLO application. This comparison demonstrates the balanced trade-off achieved by the optimized MobileNet V2 model, which maintains competitive detection accuracy while delivering the computational efficiency necessary for real-time road distress detection on resource-constrained devices.

The field validation results confirm that the MobiLiteNet framework's systematic optimization approach effectively balances

detection accuracy and computational efficiency in real-world deployment scenarios.

All detection results were automatically saved on the respective devices with timestamps. These data are scheduled for subsequent upload to a cloud database system, ensuring effective data integration for large-scale road condition monitoring and supporting real-time infrastructure management decisions. This systematic approach to data collection and management demonstrates the scalability of the proposed solution for widespread implementation in road maintenance programs, contributing to more efficient and timely infrastructure management.

## Discussion

The present study introduces MobiLiteNet, a lightweight deep learning framework optimized for real-time road monitoring, enabling efficient deployment on smartphones and MR devices. By integrating ECA, structural refinement, sparse knowledge distillation, structured pruning, and quantization, the framework significantly reduces model complexity while maintaining high accuracy. ECA enhances feature representation, knowledge distillation ensures strong generalization from high-capacity teacher models, and pruning with quantization optimizes execution efficiency on embedded platforms. These optimizations collectively bridge the gap between high-performance deep learning models and resource-constrained environments, facilitating their deployment on mobile devices. The workflow includes data collection, model optimization, mobile deployment, and field validation, forming a seamless and scalable pipeline for accurate road distress detection in real-world applications.

In the dataset construction and augmentation phase, diverse road distress images were collected from multiple geographical regions, including Aachen (Germany), Beijing (China), and Swansea (United Kingdom), capturing variations caused by different climatic conditions and traffic patterns. The use of WGAN-GP for data augmentation significantly enhanced the dataset's robustness, addressing class imbalance and improving the model's ability to generalize to unseen data. This comprehensive approach to data preparation ensured that the models were trained on a diverse set of images, contributing to their high performance during field validation.

The model optimization process demonstrated the effectiveness of the MobiLiteNet framework. The transition from server-side training to mobile deployment enabled successful integration into the RoadIntelligent application on both smartphone and MR platforms. The ablation studies confirmed each component's contribution to the framework's overall performance, validating that the combined optimization approach effectively addresses the computational constraints of mobile devices while maintaining robust detection capabilities.

The deployment across smartphone and MR platforms demonstrates the framework's versatility in diverse operational contexts. Smartphones enable accessible, widespread deployment, while MR devices provide enhanced visualization capabilities for inspectors. This dual-platform approach significantly expands the potential application scenarios for automated road distress detection in various operational environments.

Field validation conducted in Aachen, Germany, provided real-world evidence of the optimized model's robustness and generalization capabilities. The RoadIntelligent application demonstrated high detection accuracy with efficient processing speeds, enabling real-time analysis on resource-constrained devices. Comparative testing against the YOLO application revealed that while both achieved similar detection accuracy, the MobiLiteNet-optimized model processed images significantly faster than the more complex YOLO alternative. This performance advantage validates the effectiveness of the optimization techniques in operational environments and affirms the framework's potential for enhancing road maintenance operations and infrastructure management.

In conclusion, the MobiLiteNet framework provides a strong foundation for the next generation of automated infrastructure assessment systems. Its adaptability to smartphones and MR platforms supports diverse applications beyond road monitoring, such as early warning systems for visually impaired individuals and broader smart city initiatives. The fusion of mobile computing with artificial intelligence, demonstrated by advancements in AR technologies like Meta's Orion glasses, highlights the potential for enhancing real-time infrastructure monitoring and situational awareness. Despite these strengths, several limitations persist. The current implementations of RoadIntelligent and YOLO applications represent initial prototypes with basic functionality focused primarily on demonstrating technical feasibility rather than comprehensive solutions. The model's dependence on high-quality input data may limit performance in environments with poor image quality or inconsistent data. Additionally, extreme conditions such as severe weather or low-light scenarios can affect detection accuracy. Future research will aim to improve the model's robustness under adverse conditions, incorporate multi-sensor data for greater resilience, and develop more comprehensive real-world applications with enhanced user interfaces, expanded distress classification capabilities, and integration with existing infrastructure management systems. Addressing these challenges in future studies will enable MobiLiteNet to significantly contribute to the development of intelligent, inclusive, and resilient urban environments.

## Methods

### Experiment environment

The experimental setup for this study involved diverse hardware platforms to evaluate the performance of the MobiLiteNet framework. Model training and optimization were conducted on a high-performance server equipped with an NVIDIA RTX 4090 GPU (24 GB VRAM) using Python 3.10 and PyTorch 2.4. For mobile deployment, a Huawei P40 smartphone, powered by a Kirin 990 5G chipset with 8 GB RAM and Android 10, was used to assess real-time performance under typical mobile conditions. The development environment utilized Android Studio Ladybug Feature Drop for application implementation. Additionally, MR deployment experiments were carried out using the Meta Quest 3, featuring a Snapdragon XR2 Gen2 SoC and 8 GB DRAM.

### ECA for feature enhancement

ECA[39] is a lightweight attention mechanism designed to enhance channel-wise feature representation in CNNs. Unlike traditional attention mechanisms that rely on fully connected layers, ECA introduces a simple yet effective approach using a one-dimensional convolution operation without dimensionality reduction. This method allows for efficient modeling of cross-channel interactions while maintaining low computational complexity.

The core principle of ECA is to capture local cross-channel dependencies by applying a 1D convolution with an adaptive kernel size determined based on the number of channels[39]. This eliminates the need for additional parameters or complex transformations, ensuring computational efficiency. The kernel size is selected to balance the trade-off between model capacity and efficiency, enabling effective attention allocation across channels. As a result, ECA improves the model's ability to focus on informative features, enhancing performance in various computer vision tasks.

### Knowledge distillation

Knowledge distillation is a model compression technique designed to transfer knowledge from a large, high-capacity teacher model to a smaller, more efficient student model. The goal is to retain the performance of the teacher model while significantly reducing the computational complexity, making the student model suitable for deployment on devices with limited computational resources.

The core principle of knowledge distillation is to train the student model to approximate the behavior of the teacher model. Traditionally, this is achieved by minimizing a distillation loss function that incorporates both soft target and hard target losses[40]. In this work, the distillation framework is extended by introducing an additional L1 regularization term to enhance model sparsity, thereby improving generalization and facilitating model pruning[44]. L1 regularization is selected over L2 regularization for its ability to promote true sparsity by driving parameters to zero, directly supporting the subsequent pruning operations essential for mobile deployment.

The conventional distillation loss consists of two primary components: the soft target loss, which captures knowledge from the teacher model, and the hard target loss, which ensures alignment with ground-truth labels. This is formulated as[40]:

$$L_{distill} = L_{soft} + L_{hard} \tag{1}$$

The soft target loss, typically implemented using Kullback–Leibler (KL) divergence[58], measures the discrepancy between the student's output distribution $q_s$ and the softened teacher's output distribution $q_t$ at temperature $T$[40]:

$$L_{soft} = T^2 \cdot \mathrm{KL}(q_t \parallel q_s) = T^2 \cdot \sum_i q_t^i \log \frac{q_t^i}{q_s^i} \tag{2}$$

The hard target loss represents the conventional cross-entropy loss between the student model's predictions and the true ground-truth labels $y$[40]:

$$L_{hard} = -\sum_i y_i \log q_s^i \tag{3}$$

This term ensures that the student model not only mimics the teacher's predictions but also aligns with true labels, preserving classification accuracy.

To further improve model efficiency, an L1 regularization term is introduced to encourage sparsity in the model parameters. This helps reduce computational complexity and enhances pruning effectiveness in subsequent optimization stages. The L1 regularization term is formulated as[44]:

$$L_{reg} = \lambda ||\theta||_1 = \lambda \sum_j |\theta_j| \tag{4}$$

where $\lambda$ is a hyperparameter that controls the strength of regularization, with larger values encouraging greater sparsity in the model parameters.

By incorporating L1 regularization into the traditional knowledge distillation framework, the final loss function is defined as:

$$L_{distill\,new} = L_{soft} + L_{hard} + \lambda \parallel \theta \parallel_1 \tag{5}$$

By combining these three components, the distillation loss ensures that the student model not only learns from the teacher's comprehensive feature representations but also aligns with the true labels while maintaining a sparse and efficient architecture. This approach effectively balances model performance with computational efficiency, making knowledge distillation a powerful tool for model optimization.

## Structured pruning

Structured pruning is a model compression technique aimed at reducing the computational complexity and memory footprint of deep neural networks by removing entire structures[41] such as filters, channels, or layers, rather than individual, unstructured weights. Unlike unstructured pruning, which leads to sparse matrices and often requires specialized hardware for efficient inference, structured pruning maintains the dense matrix format, ensuring compatibility with standard hardware and software libraries while significantly improving inference speed and reducing model size.

In this work, channel pruning is employed as the primary structured pruning strategy. The pruning strategy is guided by the sparsity regularization introduced during the knowledge distillation process, which encourages the development of sparse representations in the model. Channel pruning focuses on eliminating less important channels in convolutional layers based on their contribution to the overall model performance[41]. This is achieved by evaluating the importance of each channel using a predefined criterion, such as the magnitude of the channel's weights or its impact on the model's output. Channels with lower significance are pruned, effectively reducing the number of computations required during inference while maintaining the model's accuracy.

The importance of channels is assessed through an iterative process, where the model undergoes fine-tuning after each pruning step to recover any potential loss in performance. By incorporating channel pruning, the model achieves a balanced trade-off between efficiency and accuracy, facilitating deployment on resource-constrained devices.

## Wasserstein Gan with gradient penalty (Wgan-Gp) for data augmentation

A GAN[59] is a generative model based on game theory that consists of two networks: a generator G, which generates synthetic samples from noisy variables, and a discriminator D, which represents the probability of deciding a given sample as real data.

The original GAN network training is unstable and very sensitive to hyperparameters, which can also lead to model collapse. Wasserstein GAN (WGAN)[60]. utilizes the distance between the probability distribution of real samples and of generated samples, rather than the discriminator-based objective function, to solve the pattern collapse problem. The loss function of WGAN is shown in Eq. (1)[60].:

$$L = \mathrm{E}_{x \sim P_g}[D(x)] - \mathrm{E}_{x \sim P_r}[D(x)] \tag{6}$$

However, it is observed that most of the weights are at critical values after the weights are clipped, i.e., the weights are −C or C in most cases, which largely limits the fitting ability of WGAN and can easily lead to the problems of gradient explosion and gradient disappearance. To solve these problems, Gulrajani et al. proposed WGAN-GP, which can improve the clipping weights by utilizing the gradient penalty. The loss function of WGAN-GP is defined as follows[50]:

$$L = \mathrm{E}_{z \sim P_z(z)} \big[ D(G_\theta(z)) \big] - \mathrm{E}_{x \sim P_r(x)}[D(x)] + \lambda \mathrm{E}_{\hat{x} \sim P_{\hat{x}}(\hat{x})} \left[ (\parallel \nabla_{\hat{x}} D_{\theta_D}(\hat{x}) \parallel_2 - 1)^2 \right] \tag{7}$$

Compared with WGAN, the training process of WGAN-GP is faster and more stable.

## Mobilenet V2 architecture

MobileNet V2, an enhancement of MobileNet V1, employs depth-wise separable convolutions, significantly reducing both the computational load and the number of parameters[48]. Furthermore, MobileNet V2 incorporates Linear Bottlenecks and Inverted Residuals, which improve its accuracy and efficiency.

The training process employed the adaptive learning rate tuning algorithm RMSProp (Root Mean Square Proposition)[61] to compute and update the network parameters, aiming to reach optimal values and minimize loss.

## Format conversion and APP development

Following optimization, the optimized MobiLiteNet-based improved MobileNet V2 model was converted to TFLite[56]. format using AI Edge Torch. The process involved loading the pre-trained PyTorch model, transferring it to a CPU environment, and generating sample inputs for validation. The model was then transformed into an edge-compatible format, ensuring numerical consistency between the original and converted outputs. Finally, the optimized model was exported as a TFLite file, enabling efficient deployment on mobile and MR devices. For the YOLOv8 model, the conversion process involved multiple steps: first, the model was exported from PyTorch format (.pt) to ONNX (Open Neural Network Exchange) format. Next, the ONNX model was converted to TensorFlow format (.pb) using an automated conversion tool, preserving the model's architecture and weights. The TensorFlow model was further optimized using TensorFlow's model transformation tools before being converted into TFLite format to enable efficient inference on resource-constrained devices.

The developed applications were implemented in Java and Kotlin within the Android Studio environment. These applications integrated the optimized TFLite models, facilitating real-time road distress detection on both Android smartphones and MR devices, while ensuring low-latency inference and seamless user interaction.

## Uploading detection results to the cloud database via API

The detection results from both smartphone and MR devices were automatically saved with timestamps. These structured datasets are scheduled for upload to cloud storage and future uploading to the German open-access database Mobilithek, which aims to support large-scale road condition monitoring and enabling comprehensive infrastructure management decisions based on temporally and spatially accurate distress information.

To facilitate this process, an API interface leveraging Google Drive's REST API was implemented, ensuring secure and efficient data transfer. The procedure involved establishing a secure connection with the Google Drive API using OAuth 2.0 credentials, which included obtaining and managing access tokens for session authentication. Detection results were compiled into a structured format, typically JSON or CSV, incorporating metadata such as timestamps, device IDs, and location coordinates. A new file was created in Google Drive to store these results. The structured detection results were then uploaded to this newly created file using a multipart upload request, efficiently handling large datasets. Upon successful upload, the API returned a confirmation response containing the file ID and a link to the uploaded file, which was logged for auditing purposes to ensure traceability.

## Data availability

The data generated in this study have been deposited in the Figshare repository database under accession code https://doi.org/10.6084/m9.figshare.28404875.v2 [62]. Additional data supporting the results are available within the manuscript and the Supplementary Information. Source data for all figures and tables are provided with this paper. Source data are provided with this paper.

## Code availability

The code for MobiLiteNet, the optimized MobileNet V2 variants, and other baseline models has been archived and is publicly available via Zenodo at https://doi.org/10.5281/zenodo.15227777 [63].

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

## Acknowledgements

The manuscript was partially refined using a large language model (Claude). The authors gratefully acknowledge the Autodl platform for providing access to server-class hardware, including NVIDIA RTX 4090. Special appreciation goes to Hongyu Shi, Zijin Xu, and Huiting Zhang for their contributions during the initial data collection and code preparation phase. The authors extend particular thanks to Professor Xue Luo from

Zhejiang University for her valuable contributions in the early stages of this work, and to Hancheng Zhang for providing insightful feedback and suggestions during the manuscript revision process. No external funding was received for this study. Maps shown in Figs. 1d and 7a are adapted from Stamen Toner under appropriate attribution requirements. Visual elements used in Figs. 1c, 6a, and 7b were adapted from Freepik under their free commercial use license (authors: juicy_fish, macrovector, studiogstock, and freepik, etc.). Part of the icons in Figs. 1a, b, and e and 5a were obtained from The Noun Project and used in accordance with their licensing terms (authors: Leonardo Henrique Martini, Alzam, Puspito, and Ali Mahmudi, etc.). A full list of third-party visual elements, creator names, license types, and source links is provided in the Third-Party Rights.

## Author contributions

Y. Hu and N.C. conceptualized the study and developed the code for the MobiLiteNet algorithm. Y. Hu, N.C., Y. Hou, and P.L. performed experiments and formal analyses. Y. Hu wrote the manuscript. N.C., Y. Hou, and P. L. revised and edited the manuscript. Y. Hou and P.L. supervised different parts of the project. B.J. developed the RoadIntelligent Application. X.L. developed the YOLO Application. All authors reviewed and edited the manuscript.

## Funding

## Competing interests

The authors declare no competing interests.
