## [Transparent Peer Review file · Nature Communications]

Lightweight Deep Learning for Real-Time Road Distress Detection on Mobile Devices

Corresponding Author: Professor Pengfei Liu

Version 0:

Reviewer comments:

Reviewer #1

(Remarks to the Author)

This paper presents a lightweight deep learning model based on MobileNetV2, optimized for real-time road damage detection on mobile and mixed reality (MR) devices.

1. Novelty of Approach: While the authors utilize MobileNetV2 with TensorFlow Lite for real-world deployment on mobile and mixed reality (MR) devices, the approach lacks novelty. The use of deep learning for road damage detection is well-explored, as seen in previous competitions, such as the IEEE Big Data 2018 road damage competition. Many papers since then have documented advancements in this area, which this paper does not seem to surpass significantly.

2. Assumptions about Mobile/MR Usage: The authors' choice to deploy their method on smartphones and MR devices is questionable. It seems unlikely that road maintenance professionals would prefer wearing MR devices for road damage detection, given the availability of faster and more accurate vehicle-mounted sensors and specialized detection equipment. This point raises concerns about the practicality of the proposed application and its real-world viability.

3. Originality of Method: The paper lacks a significant original contribution in terms of method development. While MobileNetV2 is a recognized model for lightweight mobile deployment, there are numerous advanced model compression and optimization techniques, such as pruning, quantization, and knowledge distillation, which could improve the model's performance specifically for resource-limited devices. The authors primarily apply existing techniques without contributing new methodologies or optimizations to the deep learning models or the field of mobile deployment.

(Remarks on code availability)

the code is not well organized, and it is hard for some to reuse and improve the work.

Reviewer #2

(Remarks to the Author)

The proposed study introduces an interesting and timely approach to pavement crack detection using lightweight AI models, particularly MobileNet V2 (and YOLOv8), optimized for mobile devices and Mixed Reality (MR) platforms. This solution holds significant potential for improving the efficiency and precision of road infrastructure inspections, while also offering societal benefits such as early warning systems for visually impaired individuals.

However, there are several areas where the paper could be strengthened:

1. Data Limitations: The dataset used in the study appears small, even with augmentation, which may limit the robustness of the deep learning model. The reported accuracy could be influenced by the size of the dataset, and the model's generalizability may be impacted. A larger, more diverse dataset would likely enhance the model's performance and reliability.

2. Experiment Details and Comparisons: The authors mention using the RDD2022 dataset for the Mixed Reality device deployment, but further details should be included, particularly regarding the scale of the experiments conducted. The

RDD2022 dataset has been extensively used in global Road Damage Detection Challenges, and it would be valuable to compare the proposed method against state-of-the-art solutions from these competitions.

3. Data Contribution: Since the dataset itself is one of the contributions of the paper, it would be beneficial to include a comparison of the proposed dataset with other well-known datasets, such as the German Asphalt Pavement Distress (GAPs) dataset and similar datasets from other countries. This would help position the proposed data within the broader landscape of pavement distress datasets.

4. Paper Length: The paper could be condensed to focus more on key findings and results, streamlining some sections without compromising on clarity. A more concise presentation would improve readability and impact. Some sections may be moved to Appendix if required.

(Remarks on code availability)

Reviewer #3

(Remarks to the Author)

The article is very interesting as it presents an advancement in the real-time domain using object detection deep learning techniques. However, the paper is poorly organized and should be divided into different sections to simplify the lecturer's work. For this reason I accept the paper for publication after major corrections.

(Remarks on code availability)

Version 1:

Reviewer comments:

Reviewer #1

(Remarks to the Author)

The authors have appropriately addressed most of my concerns. I have three suggestions that should be addressed before acceptance:

1. Please explicitly clarify the benefits of using this lightweight model on mobile devices.

2. The literature review could be enhanced with additional relevant references. I recommend including the following papers for completeness

For edge computing:

[1]Ale, L., Zhang, N., King, S.A. et al. Empowering generative AI through mobile edge computing. Nat Rev Electr Eng 1, 478–486 (2024). <https://doi.org/10.1038/s44287-024-00053-6>

For road damage detection:

[2]Maeda, H., Sekimoto, Y., Seto, T., Kashiyama, T. and Omata, H. (2018), Road Damage Detection and Classification Using Deep Neural Networks with Smartphone Images. Computer-Aided Civil and Infrastructure Engineering, 33: 1127-1141. <https://doi.org/10.1111/mice.12387>

[3]L. Ale, N. Zhang and L. Li, "Road Damage Detection Using RetinaNet," 2018 IEEE International Conference on Big Data (Big Data), Seattle, WA, USA, 2018, pp. 5197-5200, doi: 10.1109/BigData.2018.8622025.

3. I appreciate the effort made to revise the code. However, I remain concerned about the practical feasibility of their approach. Specifically, the manuscript emphasizes deploying code directly on smartphones as its key advantage, yet it is unclear how users would practically execute Python or conda environments (those are more likely run on the servers or desktops) on these devices. To strengthen the paper's argument, I suggest the authors consider developing a simple, real-world Android or iOS application to demonstrate the actual functionality and deployment feasibility of their model on smartphones.

(Remarks on code availability)

I highly recommend the authors provide some usage examples with jupyter notebooks and real smartphone (android or ios) apps to demonstrate it actually works on those devices,

Reviewer #2

(Remarks to the Author)

(Remarks on code availability)

Reviewer #3

(Remarks to the Author)

Based on the revised version and the authors' responses to the reviewers' comments, the author has addressed all the questions and has particularly improved the structure of the article. In this work, the authors proposed a new version of MobileNetV2 that is optimized for real-time detection of road distress. While there is still room for further improvement, considering all the enhancements made in this updated version, I accept the paper for publication.

(Remarks on code availability)

The codes are reorganized and commented to improve their comprehension and be reused by the scientific community.

Response to Reviewers' comments:

The authors would like to express their sincere gratitude to the reviewers for improving the quality of this review paper.

Reviewer #1:

This paper presents a lightweight deep learning model based on MobileNetV2, optimized for real-time road damage detection on mobile and mixed reality (MR) devices.

The reviewer's comment 1:

- Novelty of Approach: While the authors utilize MobileNetV2 with TensorFlow Lite for real-world deployment on mobile and mixed reality (MR) devices, the approach lacks novelty. The use of deep learning for road damage detection is well-explored, as seen in previous competitions, such as the IEEE Big Data 2018 road damage competition. Many papers since then have documented advancements in this area, which this paper does not seem to surpass significantly.

Response: We appreciate the reviewer's thoughtful feedback regarding the novelty of our approach. While it is true that deep learning for road damage detection has been widely explored, we believe our work introduces several novel contributions that address critical gaps in the field, particularly concerning real-world deployment, practical applicability, and model generalization.

1. Our primary contribution is the introduction of MobiLiteNet, a generalized, mobile-optimized lightweight deep learning framework that extends beyond the optimization of a single model, such as MobileNetV2. In contrast to the original approach, which focused solely on improving MobileNetV2, the revised version proposes a flexible framework applicable to a wide range of deep learning models intended for mobile deployment. MobiLiteNet integrates advanced optimization techniques, including Efficient Channel Attention (ECA), sparse knowledge distillation, structured pruning, and quantization, to enhance model efficiency and accuracy across different architectures. While many existing studies focus on achieving high accuracy on server-based systems with abundant computational resources, our framework is specifically designed to enable the deployment of deep learning models on resource-constrained devices, such as smartphones and Mixed Reality (MR) devices. This shift towards generalized mobile deployment represents a novel direction in road monitoring research, particularly as MR technology remains underexplored in the context of real-time infrastructure inspection and immersive field applications.
2. The integration of MR devices and smartphones into pavement detection workflows introduces a unique contribution to the field. Unlike traditional methods that rely on vehicle-mounted systems or cloud-based detection, MR devices enable hands-free, real-time visualization of pavement conditions, enhancing the efficiency and accuracy of on-site inspections. We successfully deployed our framework in real-world road engineering

projects in Aachen, Germany, demonstrating the practical feasibility of MR-based real-time monitoring in complex environments. This capability opens new opportunities for immersive training, augmented field inspections, and collaborative infrastructure management, areas that have not been comprehensively addressed in previous studies.

3. While previous studies have primarily focused on improving model performance in server-based environments, real-world applications introduce unique challenges that are often overlooked. Smartphones and MR platforms have significantly limited computational resources compared to servers, making it difficult to deploy complex deep learning models without substantial performance degradation. Our framework addresses this by integrating optimization techniques to ensure high accuracy with low computational overhead. Additionally, many existing solutions struggle with real-time processing due to computational bottlenecks, but MobiLiteNet is specifically optimized for low-latency inference, ensuring real-time performance essential for practical deployment in dynamic, on-site environments. Furthermore, current distress detection systems predominantly rely on vehicle-mounted hardware, which is expensive, rigid, and heavily dependent on specialized equipment, or cloud-based methods that require uploading terabytes of data per inspection session, leading to significant challenges in data transmission, storage, and privacy. In contrast, our approach provides a low-cost, lightweight, and portable solution, enabling flexible deployment on smartphones and MR devices, even in hard-to-reach areas where traditional methods are impractical.
4. A critical limitation of many previous studies is their reliance on standardized, publicly available datasets, which often lack the diversity and complexity of real-world scenarios. To enhance the robustness and generalization capability of our models, we constructed a diverse dataset comprising pavement distress images collected from Aachen, Germany; Beijing, China; and Swansea, UK. This dataset captures a wide range of environmental conditions (e.g., rainy, nighttime, and overcast scenarios), road types, and distress patterns unique to different regions, such as top-down cracks in German pavements and reflection cracks in Chinese roads. By incorporating such diversity, our framework achieves better generalization and robust performance across varied real-world conditions, surpassing the limitations of many existing approaches that focus on narrow, homogeneous datasets.
5. Our approach not only advances road distress detection but also extends to broader applications in infrastructure management and smart city development. Its portability and cost-effectiveness enable flexible municipal inspections and routine maintenance without specialized equipment. The framework complements existing vehicle-mounted and cloud-based systems, with MR devices enhancing real-time visualization and smartphone-based detection serving as a rapid pre-screening tool. Beyond road monitoring, MobiLiteNet supports applications like early warning systems for visually impaired individuals and contributes to smart city initiatives, leveraging advancements in AR technologies (e.g., Meta's Orion glasses) to enhance real-time infrastructure monitoring and situational awareness.

In summary, while our study builds on established deep learning techniques, its novelty lies in the practical deployment of an optimized framework for real-time, on-device road distress detection on smartphones and MR devices. By addressing critical challenges in computational constraints, real-time processing, and deployment flexibility, and by leveraging a diverse, real-world dataset, we believe our research represents a substantial advancement over existing work. The successful integration of MR technology opens new avenues for immersive, interactive infrastructure monitoring, further distinguishing our contributions from prior studies.

The reviewer's comment 2:

- Assumptions about Mobile/MR Usage: The authors' choice to deploy their method on smartphones and MR devices is questionable. It seems unlikely that road maintenance professionals would prefer wearing MR devices for road damage detection, given the availability of faster and more accurate vehicle-mounted sensors and specialized detection equipment. This point raises concerns about the practicality of the proposed application and its real-world viability.

Response: Thanks for the reviewer's comment. While vehicle-mounted sensors and specialized detection equipment are widely used in road maintenance, these methods face significant challenges that our approach seeks to address. Vehicle-mounted systems generate terabytes of data per inspection session, especially during long highway inspections, creating data transmission and storage bottlenecks. Additionally, they require complex, expensive hardware setups that limit their flexibility for localized or small-scale inspections. Cloud-based methods face similar issues, with the added challenge of real-time data processing delays due to transmission dependencies. In contrast, our approach enables on-device, real-time processing on smartphones and MR devices, significantly reducing the burden of data transfer and post-processing.

Deploying pavement detection on smartphones and MR devices offers several distinct advantages. These devices are low-cost, lightweight, and widely accessible, allowing for flexible deployment in environments where vehicle-mounted systems are impractical, such as pedestrian zones, narrow streets, or remote rural areas. Their portability makes them ideal for small maintenance teams, emergency response scenarios, and localized inspections that require quick, on-the-spot assessments. Furthermore, MR devices provide a hands-free, immersive experience for inspectors, enhancing situational awareness and real-time visualization of road conditions, while smartphones offer a familiar, user-friendly interface that lowers the barrier to adoption for non-specialists.

Our approach is designed to complement existing detection systems rather than replace them. While vehicle-mounted equipment remains suitable for large-scale, high-speed inspections, smartphones and MR devices excel in localized assessments, quick follow-ups, and detailed verifications of defects identified during broader surveys. This integration enhances the overall efficiency of road maintenance workflows by combining the strengths of both mobile and vehicle-based detection methods.

To validate the practical feasibility of our approach, we conducted real-world deployments in road engineering projects in Aachen, Germany, using MR devices and smartphones for on-site monitoring. These deployments demonstrated the effectiveness of MR technology in complex field environments, while smartphone-based validation across diverse geographies and weather conditions yielded a high accuracy, underscoring the robustness of our framework in practical scenarios.

Beyond road maintenance, our framework supports broader applications in smart city initiatives and public infrastructure monitoring. It can be adapted for early warning systems for visually impaired individuals or integrated into municipal apps reporting of road conditions. The fusion of mobile computing with artificial intelligence, as demonstrated by emerging AR technologies like Meta's Orion glasses, highlights the potential for enhancing real-time infrastructure monitoring and situational awareness in urban environments.

While we recognize that MR devices may not be suitable for all scenarios, especially high-speed or extensive highway inspections, their portability, flexibility, and real-time capabilities offer valuable advantages in specific use cases. As MR technology continues to advance, with improvements in battery life, processing power, and wearability, we anticipate even broader adoption in infrastructure monitoring. Future work will focus on enhancing multi-sensor integration and optimizing real-time processing to further extend the applicability of smartphones and MR-based detection systems.

The reviewer's comment 3:

- Originality of Method: The paper lacks a significant original contribution in terms of method development. While MobileNetV2 is a recognized model for lightweight mobile deployment, there are numerous advanced model compression and optimization techniques, such as pruning, quantization, and knowledge distillation, which could improve the model's performance specifically for resource-limited devices. The authors primarily apply existing techniques without contributing new methodologies or optimizations to the deep learning models or the field of mobile deployment.

Response: Thanks for the reviewer's insightful comments regarding the originality of our method, which prompted us to enhance our work in the revised manuscript. While the original version focused on optimizing MobileNetV2, we have expanded our approach to propose MobiLiteNet, a generalized, integrated framework that systematically combines techniques such as pruning, quantization, and knowledge distillation. Unlike prior studies limited to specific models, MobiLiteNet is model-agnostic, enabling the efficient, scalable deployment of various deep learning architectures on smartphones and MR devices, representing a significant advancement in mobile deep learning deployment.

The novelty of MobiLiteNet lies in the strategic integration and tailoring of established optimization techniques specifically for mobile deployment. In the revised manuscript, the Results section titled "The Proposed MobiLiteNet Framework" illustrates the entire optimization process and its impact on the model architecture, as shown in Figure 2.

Additionally, the section “Model Optimization and Performance Evaluation” provides a detailed comparison between the original model and the framework-optimized version, highlighting differences in architecture, parameter count, and accuracy, as depicted in Figure 4.

Fig.2 | Architecture Optimization in the MobiLiteNet Framework. a Process Schematic for MobiLiteNet Architecture. **b** Architecture and Implementation Details Based on MobileNet V2.

Fig. 4 | Structural Optimization of the Baseline Model Using MobiLiteNet Framework. a Original structure of MobileNet V2. **b** Optimized model with MobiLiteNet framework.

For example, sparse knowledge distillation is combined with L1 regularization to promote parameter sparsity, which enhances the effectiveness of structured pruning in reducing model complexity while maintaining high accuracy. The integration of Efficient Channel Attention (ECA) further improves the model’s ability to focus on critical features with minimal

computational overhead, ensuring real-time performance on mobile devices. The quantization process converts model weights from float32 to INT8, resulting in significant performance improvements across different platforms: on smartphones, inference speed increased by 3.5x (from 122.3 ms to 34.5 ms per image), and on MR devices, by 7.2x (from 288.9 ms to 40.1 ms per image), while maintaining robust detection accuracy of 96.38%. These results demonstrate the practical effectiveness of our approach in optimizing deep learning models for resource-constrained smartphone and MR environments.

To validate the contributions of each component within the MobiLiteNet framework, we conducted extensive ablation studies, detailed in the revised manuscript under "MobiLiteNet Deployment Workflow and Component-wise Ablation Analysis". These experiments isolate and quantify the impact of each optimization technique on model performance, demonstrating that each component plays a critical role in enhancing accuracy, model size, and computational efficiency. As shown in Figure 5b and 5c, each progressive optimization step contributes meaningfully to the framework's overall effectiveness. Furthermore, to evaluate practical performance across different hardware platforms, we measured end-to-end processing times for each model variant on both smartphones and MR devices, as presented in Figure 6b. These measurements reveal significant efficiency improvements with each optimization stage, with the fully optimized model achieving processing times of 34.5 ms on smartphones and 40.1 ms on MR devices—representing 71.8% and 86.1% reductions compared to the baseline MobileNet V2, respectively. This systematic evaluation confirms the critical role of each component in the framework's integrated design and validates its practical efficiency for real-time applications on resource-constrained devices.

Fig. 5 | End-to-End Deployment and Ablation Study of MobiLiteNet Framework. **a** Process flow for mobile deployment and inference. **b** Ablation study of model optimization in MobiLiteNet framework. **c** Confusion matrices for ablation study models: BR: broken pavement marking. LO: longitudinal crack. MA: alligator crack. TR: transverse crack.

a Smartphones and Mixed Reality Devices

b End-to-End Processing Time Comparison of Different Models on Mobile Devices

	ResNet	MobileNet V2	Model 1	Model 2	Model 3	Model 4	Model 5
Speed of Smartphones (ms)	592.6	122.3	122.5	71.5	70.7	47.2	34.5
Speed of MR Devices (ms)	1450.4	288.9	289.9	129.6	128.5	64.8	40.1

Fig. 6 | Practical Deployment and Performance Comparison on Mobile Devices. **a** Smartphone-based and MR-based application deployment visualization. **b** End-to-end processing time comparison of different models on mobile devices.

In addition to methodological advancements, the practical novelty of our work lies in the real-world deployment of optimized deep learning models on smartphones and MR devices for real-time road monitoring. Unlike existing research that focuses primarily on server-based accuracy improvements, our framework addresses the real-world challenges of limited computational resources, battery constraints, and latency that come with mobile deployment. This was demonstrated through successful deployments in road engineering projects in Aachen, Germany, where our models achieved real-time, on-device detection with high accuracy and efficiency, confirming the framework’s applicability in complex field environments.

Furthermore, the broader applicability and scalability of MobiLiteNet extend beyond road distress detection. The framework is adaptable to various infrastructure monitoring tasks and supports applications in smart city initiatives, such as early warning systems for visually impaired individuals. The ability to generalize this framework across different models and real-world use cases highlights its original contribution to the field of mobile deep learning deployment, offering a practical solution that addresses both technical optimization and real-world feasibility.

In summary, our contribution lies in the development of MobiLiteNet, a generalized framework for deploying deep learning models on resource-constrained smartphones and MR devices. By systematically integrating optimization techniques and validating them through ablation studies and real-world deployments, our work represents a significant advancement in mobile deep learning and infrastructure monitoring.

The reviewer’s comment 4 (Remarks on code availability):

- The code is not well organized, and it is hard for some to reuse and improve the work.

Response: Thank you for highlighting the code organization issue. We have rewritten and restructured all the codes with the corresponding information in the revised submission, adding detailed comments and documentation to improve its readability, reusability, and ease of modification for future work.

Reviewer #2:

The proposed study introduces an interesting and timely approach to pavement crack detection using lightweight AI models, particularly MobileNet V2 (and YOLOv8), optimized for mobile devices and Mixed Reality (MR) platforms. This solution holds significant potential for improving the efficiency and precision of road infrastructure inspections, while also offering societal benefits such as early warning systems for visually impaired individuals.

The reviewer's comment 1:

- Data Limitations: The dataset used in the study appears small, even with augmentation, which may limit the robustness of the deep learning model. The reported accuracy could be influenced by the size of the dataset, and the model's generalizability may be impacted. A larger, more diverse dataset would likely enhance the model's performance and reliability.

Response:

Thanks very much for the reviewer's recognition of our work!

We appreciate the reviewer's valuable feedback regarding the dataset size and its potential impact on the model's robustness and generalizability. In response to this concern, we have significantly expanded both the size and diversity of our dataset in the revised manuscript to enhance the model's robustness, generalization, and reliability.

As detailed in the "Dataset Construction and Augmentation" section, we have substantially expanded our dataset to include 2,873 original images collected from diverse locations including Aachen (Germany), Beijing (China), and Swansea (UK), as illustrated in Figure 3. This geographic diversity captures important variations in pavement distress characteristics due to differences in temperature variations, moisture exposure, and traffic loads. For instance, Aachen experiences moderate seasonal temperature variations and high traffic intensity, leading to distress patterns influenced by both thermal fluctuations and heavy dynamic loads. Beijing, with its extreme seasonal temperature shifts, undergoes significant thermal expansion and contraction cycles, impacting pavement durability. Swansea's high humidity and frequent rainfall lead to persistent moisture exposure, making water-related deterioration mechanisms more prevalent.

To account for a wider range of environmental conditions, the dataset was expanded to include images captured during rainy weather, at night, and from various viewing angles (both parallel and oblique perspectives), ensuring model robustness across different lighting conditions and camera positions. This diversity in environmental conditions and viewing angles significantly enhances the model's ability to generalize to real-world scenarios where lighting and camera positioning vary considerably.

The collected dataset was further augmented using a Wasserstein Generative Adversarial Network with Gradient Penalty (WGAN-GP), generating high-quality synthetic images for

each distress category. Through this augmentation process, we generated 18,000 synthetic images, substantially increasing the overall dataset size and diversity. This augmentation approach not only increased the dataset size but also enhanced its diversity and addressed potential class imbalance issues, further improving the model's generalization capabilities.

Fig. 3 | Summary of Dataset Composition and Augmentation. **a** Collected and generated images of road distresses. Categorized into four types: longitudinal cracks, transverse cracks, alligator cracks, and broken pavement markings. Each category includes images captured under four conditions: (i) parallel view, (ii) oblique view, (iii) rainy weather, and (iv) nighttime. **b** Dataset used in experiments. **c** Comparison of experimental results for the MobileNet family models.

To validate the model's robustness in real-world conditions, we conducted field validations in Aachen, Germany, as described in the "Field Validation Using Smartphones and Mixed Reality Devices" section. For smartphone-based validation, 400 road distress images were collected across various locations, with the model achieving an overall accuracy of 92.5%. Similarly, MR-based validation using 200 high-resolution images demonstrated 91% accuracy with fast detection times (87 ms). These results confirm the model's ability to generalize to real-world conditions despite variations in lighting, camera angles, and environmental factors.

These comprehensive improvements to the dataset size, diversity, and validation methodology have significantly enhanced the robustness and generalization capability of our models, addressing the reviewer's concerns about data limitations while demonstrating the practical effectiveness of our approach in diverse real-world scenarios.

The reviewer's comment 2:

- Experiment Details and Comparisons: The authors mention using the RDD2022 dataset for the Mixed Reality device deployment, but further details should be included, particularly regarding the scale of the experiments conducted. The RDD2022 dataset has been extensively used in global Road Damage Detection Challenges, and it would be valuable to compare the proposed method against state-of-the-art solutions from these competitions.

Response: Thank you for the reviewer's insightful comments regarding the experimental details and use of the RDD2022 dataset. To clarify, while RDD2022 was incorporated in our study, it was not the primary focus of our research. Our work primarily addresses the practical challenges of deploying efficient deep learning models on resource-constrained smartphones and Mixed Reality (MR) devices—a dimension that extends beyond the conventional accuracy metrics typically emphasized in global Road Damage Detection Challenges. The key contribution of our research lies in developing the MobiLiteNet framework, which systematically optimizes neural networks for real-time performance while maintaining high detection accuracy in resource-limited environments.

In our study, the YOLO-based object detection model was trained using the RDD2022 dataset, but this component represents only a small part of our overall methodology. Due to the specific constraints of MR device deployment, we employed a hybrid dataset combining RDD2022 with images collected from MR devices to enhance generalization. We acknowledge that our detection performance may not surpass the state-of-the-art results from specialized competitions, as those models are often optimized for accuracy using powerful server-based systems with fewer computational limitations.

In the revised manuscript, we addressed this by introducing MobiLiteNet, an integrated, model-agnostic optimization framework designed to enable the efficient deployment of deep learning models on resource-constrained smartphones and MR devices. Unlike the original manuscript, which focused on optimizing a single model, MobileNetV2, the revised version proposes a generalized framework that systematically integrates techniques such as Efficient Channel Attention (ECA), sparse knowledge distillation, structured pruning, and quantization. This framework ensures a balance between accuracy, model size, and inference speed, facilitating real-time deployment on mobile platforms.

Moreover, a key contribution of our work is the creation of a new, diverse dataset collected from Aachen (Germany), Beijing (China), and Swansea (UK), under various environmental conditions and viewing angles. This dataset enhances the model's robustness and generalization beyond what is achievable with RDD2022 alone. The expanded dataset and the optimization techniques introduced in MobiLiteNet were validated through field deployments on both smartphones and MR devices, demonstrating the framework's effectiveness in practical applications.

In summary, while our object detection results may not exceed those from specialized competitions using RDD2022, the core contribution of our work lies in the development of the MobiLiteNet framework and its ability to efficiently deploy deep learning models on smartphones and MR devices. The integration of a new, diverse dataset and the framework's application in real-world scenarios highlight the practical advancements our work brings to mobile deep learning deployment and infrastructure monitoring.

The reviewer's comment 3:

- Data Contribution: Since the dataset itself is one of the contributions of the paper, it would be beneficial to include a comparison of the proposed dataset with other well-known datasets, such as the German Asphalt Pavement Distress (GAPs) dataset and similar datasets from other countries. This would help position the proposed data within the broader landscape of pavement distress datasets.

Response: We would like to thank the reviewer for highlighting the importance of situating our dataset within the broader context of existing pavement distress datasets. In response, we have expanded and diversified our dataset and provided a detailed comparison with well-known public datasets such as German Asphalt Pavement Distress (GAPs), CRACK500, and CrackForest. This comparison underscores the unique contributions of our dataset, particularly in terms of geographical diversity, environmental variability, and practical applicability for deployment on smartphones and Mixed Reality (MR) devices.

In the revised manuscript, the section “Dataset Construction and Augmentation” provides a comprehensive overview of our dataset’s composition and characteristics. Our dataset consists of 2,873 original RGB images with a resolution of 512×512 pixels, which were augmented to create a total of 18,000 images. The images were collected from Aachen, Germany; Beijing, China; and Swansea, UK, capturing a wide range of pavement distress patterns, environmental conditions (such as cloudy, rainy, and nighttime scenarios), and camera angles (including oblique and parallel perspectives). The dataset is categorized into four types of pavement distress, ensuring comprehensive coverage of various real-world conditions.

To position our dataset within the broader context, we compared it to GAPs, CRACK500, and CrackForest:

- GAPs provides high-quality images of asphalt pavement distress from Germany but focuses primarily on specific distress types under controlled conditions. In contrast, our dataset exhibits greater diversity, including data from Germany, China, and the UK, capturing a wider range of environmental conditions such as varying climates and temperatures that influence pavement distress formation. This multi-regional approach accounts for different pavement materials and construction practices, offering a more comprehensive dataset for global applications and enhancing the model’s ability to generalize across diverse real-world scenarios.
- While CRACK500 offers a large collection of crack images, it is limited to crack detection and does not encompass the broader range of pavement distress categories present in our dataset. Furthermore, CRACK500 primarily features images taken under uniform lighting conditions, whereas our dataset includes images captured in adverse weather and low-light environments, which are crucial for models intended for real-world deployment.
- CrackForest is recognized for its detailed annotations but is geographically limited and lacks the environmental diversity and multi-angle perspectives present in our dataset. Furthermore, the images in CrackForest are relatively small in size (480×320 pixels), which may restrict their effectiveness for training models that require more spatial detail. In comparison, our dataset includes 512×512 pixel RGB images collected from Germany, China, and the UK, encompassing a wide range of environmental conditions and camera

angles, offering a more comprehensive training set for models that need to generalize across diverse global road conditions.

-

Additionally, our dataset emphasizes practical applicability. Images were collected using smartphones and MR devices, reflecting real-world data acquisition methods aligned with our proposed framework's deployment environments. This ensures that models trained on our dataset are optimized for real-time performance on smartphones and MR devices, addressing challenges that datasets like GAPS, CRACK500, and CrackForest do not directly address.

REFERENCES

44. Eisenbach, M., Stricker, R., Seichter, D. et al. How to get pavement distress detection ready for deep learning? A systematic approach. *Proc. 2017 Int. Joint Conf. Neural Netw. (IJCNN)*, 2039-2047 (2017).

45. Stricker, R. et al.. Road Surface Segmentation - Pixel-Perfect Distress and Object Detection for Road Assessment. *Proc. Int. Conf. Autom. Sci. Eng. (CASE)*, 1-8 (2021).

46. Stricker, R. et al.. Improving Visual Road Condition Assessment by Extensive Experiments on the Extended GAPS Dataset. *Int. Joint Conf. on Neural Networks (IJCNN)*, 1-8 (2019).

47. Yang, F., Zhang, L., Yu, S., et al. Feature pyramid and hierarchical boosting network for pavement crack detection. *IEEE Trans. Intell. Transp. Syst.* 21(4), 1525-1535 (2019).

48. Shi, Y., Cui, L., Qi, Z., et al. Automatic road crack detection using random structured forests. *IEEE Trans. Intell. Transp. Syst.* 17(12), 3434-3445 (2016).

The reviewer's comment 4:

- Paper Length: The paper could be condensed to focus more on key findings and results, streamlining some sections without compromising on clarity. A more concise presentation would improve readability and impact. Some sections may be moved to Appendix if required.

Response: Thanks for the reviewer's comment. In the revised manuscript, we have made significant adjustments to enhance both logical flow and clarity while maintaining the focus on key findings and results.

To improve the manuscript's structure, we have reorganized several sections to create a more concise narrative, emphasizing the critical aspects of the MobiLiteNet framework and its performance without compromising on essential details. Additionally, we have moved the technical details related to YOLO model training and deployment to supplementary materials, allowing readers to focus on the main contributions of the paper while still having access to the complete methodological information.

Furthermore, at the end of the "Workflow of Experimental and Results Overview" in the Results section, we have added a concluding paragraph that provides a clear roadmap of the subsequent sections. This addition enhances readability by helping readers navigate through the progression of experiments and findings more effectively.

To provide a comprehensive understanding of the real-world implementation, we have also

included two supplementary video files demonstrating the deployment of our framework on smartphone and MR devices. These visual demonstrations complement the written description and offer readers a clearer perspective on the practical applications of our work.

Reviewer #3:

The article is very interesting as it presents an advancement in the real-time domain using object detection deep learning techniques. However, the paper is poorly organized and should be divided into different sections to simplify the lecturer's work. For this reason I accept the paper for publication after major corrections.

The reviewer's comment 1:

- To maintain real-time application, why authors don't use quantization techniques. It's important to discuss this point in the paper.

Response: We appreciate the reviewer's insightful suggestion regarding the use of quantization techniques to enhance real-time performance. In the revised manuscript, we have expanded our approach to introduce MobiLiteNet, a generalized optimization framework designed for efficient deployment of deep learning models on smartphones and Mixed Reality (MR) devices. As part of this framework, quantization has been fully integrated as a key component, as illustrated in Figure 2.

Fig.2 | Architecture Optimization in the MobiLiteNet Framework. a Process Schematic for MobiLiteNet Architecture. **b** Architecture and Implementation Details Based on MobileNet V2.

The MobiLiteNet framework systematically incorporates five optimization techniques, including quantization as the final step in the pipeline. As detailed in the "The Proposed MobiLiteNet Framework" section, after applying Efficient Channel Attention (ECA), structural refinement, sparse knowledge distillation, and structured pruning, the model implements quantization, converting floating-point operations (float32) to 8-bit integer (INT8) representations. This conversion reduces the model size to approximately one-fourth of its original size, significantly decreasing memory usage while enhancing processing efficiency through faster arithmetic operations optimized for mobile hardware.

To evaluate the impact of quantization and other optimization techniques, comprehensive ablation studies were conducted, as presented in the "MobiLiteNet Deployment Workflow and Component-wise Ablation Analysis" section. These studies demonstrate that quantization significantly contributes to the model's efficiency without compromising accuracy. The final model incorporating quantization (Model 5) achieves a model size of merely 0.6 MB while maintaining a robust accuracy of $96.38 \pm 0.15\%$. Furthermore, processing time measurements reveal remarkable efficiency gains: on smartphones, the average inference time decreased from 122.3 ms to 34.5 ms per image, and on MR devices, from 288.9 ms to 40.1 ms per image, representing 3.5x and 7.2x increases in speed respectively.

Fig. 5 | End-to-End Deployment and Ablation Study of MobiLiteNet Framework. **a** Process flow for mobile deployment and inference. **b** Ablation study of model optimization in MobiLiteNet framework. **c** Confusion matrices for ablation study models: BR: broken pavement marking. LO: longitudinal crack. MA: alligator crack. TR: transverse crack.

We are grateful for the reviewer’s suggestion, which has helped us enhance both the technical depth and practical applicability of our framework. The inclusion of quantization within the MobiLiteNet framework represents a significant improvement in our approach, ensuring that the models are both lightweight and high performing for real-time applications.

The reviewer’s comment 2:

- The authors are invited to add a paragraph that introduce the different sections of the paper.

Response: We have made modifications in the Results section to enhance the readability and logical flow of the manuscript.

Specifically, in the first subsection of the Results, titled “Workflow of Experimental and Results

Overview”, we added a paragraph that provides a clear roadmap of the subsequent sections. This addition helps guide readers through the progression of our experiments and findings, improving the overall clarity of the paper. The paragraph reads as follows:

The subsequent sections present detailed experimental results, beginning with an in-depth analysis of the MobiLiteNet's architecture and optimization techniques. The performance of the proposed model is compared against baseline model (original MobileNet V2), with a focus on three key metrics: detection accuracy, model architecture, and parameter size. This comparison aims to demonstrate how the architectural improvement and optimizations within MobiLiteNet contribute to the improved efficiency and accuracy. An ablation study was then conducted to analyze the contributions of each component to the overall performance, identifying the specific impact of individual techniques incorporated in MobiLiteNet. Following this systematic evaluation, deployment results on smartphones and MR devices are discussed to illustrate the framework's generalization capabilities across different environments. Finally, the effectiveness of the system was validated through on-site experiments, showcasing its practical applicability in real-world road monitoring scenarios.

This structural enhancement ensures that readers have a clear understanding of the paper’s organization, thereby improving its readability and impact. We thank the reviewer for this valuable recommendation, which has helped us present our work in a more accessible and coherent manner.

The reviewer’s comment 3:

- After the introduction, the authors should explain their contribution on the proposed models Mobilenet2 and YOLOv8n in a new section.

Response: Thanks for the reviewer’s comment. We have revised the manuscript to better articulate our contributions related to MobileNet V2 optimization and the broader MobiLiteNet framework.

In the revised manuscript, we have added a dedicated contributions paragraph at the end of the Introduction section that clearly outlines the key innovations and practical advancements of our work. This paragraph reads:

The main contributions of this study are as follows:

- 1. The novel MobiLiteNet framework is proposed, integrating advanced deep learning optimization techniques to achieve efficient and accurate road distress detection suitable for smartphones and MR devices.*
- 2. A diverse dataset is constructed, consisting of road distress images collected from representative regions in Europe and Asia, capturing a wide range of real service conditions, thereby enhancing the robustness and generalization capability of the trained models.*
- 3. The optimized MobileNet V2 model, developed through the MobiLiteNet framework and trained on the diverse dataset, demonstrates effective performance in field deployment on smartphone and MR device in Aachen, Germany, validating its computational efficiency and*

detection accuracy for real-time road monitoring in complex environments.

Furthermore, in the Results section, we have added a dedicated subsection titled "The Proposed MobiLiteNet Framework" that provides a comprehensive explanation of the framework and its application to MobileNet V2. This section details how MobiLiteNet integrates five key components—Efficient Channel Attention (ECA), structural refinement, sparse knowledge distillation, structured pruning, and quantization—to systematically optimize neural network architectures for mobile deployment.

These additions clarify the role of MobileNet V2 within the broader context of our proposed framework and highlight how our contributions extend beyond optimizing a single architecture to providing a scalable solution for deploying various deep learning models on resource-constrained devices.

The reviewer's comment 4:

- It's difficult to read this paper as presented. The whole paper and sections of the paper should be reorganized and devised.

Response: We appreciate the reviewer's feedback regarding the organization and readability of the paper. In response, we have conducted a comprehensive reorganization of the manuscript to improve its clarity, logical flow, and overall readability.

The paper's structure has been significantly refined to emphasize the key contributions and to guide the reader more effectively through each section. Specifically, we have:

1. Revised the Introduction to clearly outline the scope of the study and highlight the main contributions at the end of the section, providing a strong foundation for the rest of the paper.
2. Added a new section titled "The Proposed MobiLiteNet Framework" that comprehensively explains the optimization approach and its application to MobileNet V2, illustrated through clear diagrams in Figure 2.
3. Reorganized the Results section to follow a more logical progression, beginning with dataset construction, followed by model optimization, ablation studies, cross-platform deployment analysis, and field validation.
4. Added a concluding paragraph in the "Workflow of Experimental and Results Overview" subsection that provides readers with a clear roadmap of the subsequent results, improving the navigability of the experimental findings.
5. Enhanced transitions between sections to create a more coherent narrative, allowing readers to follow the introduction, methodology, results, and discussions more easily.
6. Moved technical details regarding YOLO model training and deployment to supplementary

materials to maintain focus on the primary contributions while ensuring all methodological information remains accessible.

7. Included two supplementary video files demonstrating the practical application of our framework on smartphone and MR devices, enhancing the comprehensibility of our real-world implementations.

These structural improvements have significantly enhanced the readability of the manuscript. The revised organization provides a clearer progression from the theoretical framework to practical implementation and validation, making the paper more accessible and engaging for readers. We are grateful for this suggestion, which has contributed to making the paper more accessible and coherent.

Response to Reviewers' comments:

Thank you for your constructive feedback on our manuscript "Lightweight Deep Learning for Real-Time Road Distress Detection on Mobile Devices." We appreciate the time and expertise dedicated to reviewing our work. We have carefully addressed comments and suggestions, which have significantly improved the quality of our manuscript.

Reviewer #1:

Remarks to the Author:

The authors have appropriately addressed most of my concerns. I have three suggestions that should be addressed before acceptance:

1. Please explicitly clarify the benefits of using this lightweight model on mobile devices.

Response: Thank you for this valuable suggestion which has helped us strengthen the clarity and impact of our manuscript.

We recognized the need to more explicitly articulate the practical advantages of lightweight approach in mobile environments. The benefits of using lightweight models extend beyond theoretical improvements in model architecture to tangible advantages in field operations. These include faster processing times that enable real-time detection, reduced storage requirements that accommodate diverse hardware configurations, extended battery life during field operations, and the ability to run concurrently with other applications without performance degradation. Furthermore, the efficiency improvements achieved through our MobiLiteNet framework directly enhance the accessibility and scalability of automated road monitoring solutions, contributing to more cost-effective infrastructure management.

We have added clarification regarding these benefits in the "Cross-Platform Deployment and Performance Analysis on Smartphones and MR Devices" section:

The MobiLiteNet-optimized model delivers several key benefits for mobile-based road distress detection. The reduced processing time (34.5 ms on smartphones, 40.1 ms on MR devices) enables real-time detection at operational speeds critical for efficient field inspections. The smaller model size (from 8.9 MB to 0.6 MB) allows deployment across diverse hardware configurations while minimizing storage requirements. These optimizations extend device battery life during field operations and enable concurrent execution with other applications without performance degradation. The significant improvements in processing efficiency (71.8% reduction on smartphones, 86.1% on MR devices) directly translate to reduced operational costs, enhanced accessibility, and improved scalability for practical road monitoring implementations.

2. The literature review could be enhanced with additional relevant references. I recommend including the following papers for completeness

For edge computing:

[1]Ale, L., Zhang, N., King, S.A. et al. Empowering generative AI through mobile edge computing. *Nat Rev Electr Eng* 1, 478–486 (2024). <https://doi.org/10.1038/s44287-024-00053-6>

For road damage detection:

[2]Maeda, H., Sekimoto, Y., Seto, T., Kashiyama, T. and Omata, H. (2018), Road Damage Detection and Classification Using Deep Neural Networks with Smartphone Images. *Computer-Aided Civil and Infrastructure Engineering*, 33: 1127-1141. <https://doi.org/10.1111/mice.12387>

[3]L. Ale, N. Zhang and L. Li, "Road Damage Detection Using RetinaNet," 2018 IEEE International Conference on Big Data (Big Data), Seattle, WA, USA, 2018, pp. 5197-5200, doi: 10.1109/BigData.2018.8622025.

Response: Thank you for suggesting these valuable references.

These publications significantly contribute to the fields of edge computing and road damage detection using deep learning approaches. Integrating these works strengthens the foundation of our study by acknowledging recent advancements in mobile computing technologies for infrastructure monitoring and establishes a more comprehensive background on the evolution of deep learning methods for road distress detection.

REFERENCES

17. Ale, L., Zhang, N. & Li, L. Road damage detection using RetinaNet. In Proc. IEEE Int. Conf. Big Data, 5197–5200 (2018).

18. Maeda, H., Sekimoto, Y., Seto, T., et al. Road damage detection and classification using deep neural networks with smartphone images. *Comput. Aided Civ. Infrastruct. Eng.* 33, 1127–1141 (2018).

32. Ale, L., Zhang, N., King, S. A., et al. Empowering generative AI through mobile edge computing. *Nat. Rev. Electr. Eng.* 1, 478–486 (2024).

33. Ale, L., Zhang, N., Fang, X., et al. Delay-aware and energy-efficient computation offloading in mobile-edge computing using deep reinforcement learning. *IEEE Trans. Cogn. Commun. Netw.* 7, 881–892 (2021).

3. I appreciate the effort made to revise the code. However, I remain concerned about the practical feasibility of their approach. Specifically, the manuscript emphasizes deploying code directly on smartphones as its key advantage, yet it is unclear how users would practically execute Python or conda environments (those are more likely run on the servers or desktops) on these devices. To strengthen the paper's argument, I suggest the authors consider developing a simple, real-world Android or iOS application to demonstrate the actual functionality and deployment feasibility of their model on smartphones.

Response: Thank you for this insightful comment regarding the practical implementation aspects of our approach.

The reviewer raises a valid concern about the practical deployment of deep learning models on

mobile platforms, particularly regarding the execution environment. While our research demonstrates the theoretical feasibility and performance benefits of lightweight models on mobile devices, we acknowledge that the current implementation has limitations in terms of user accessibility and deployment readiness.

We are committed to developing a more comprehensive and user-friendly application in future work. This will include creating a fully functional Android application with an intuitive interface that enables direct deployment of our models without requiring users to manage Python environments or other technical configurations.

We have addressed this limitation in the "Discussion" section by explicitly acknowledging the current state of our applications and outlining our plans for future development. The revised text now includes:

The current implementations of RoadIntelligent and YOLO applications represent initial prototypes with basic functionality focused primarily on demonstrating technical feasibility rather than comprehensive solutions.

Future research will aim to improve the model's robustness under adverse conditions, incorporate multi-sensor data for greater resilience, and develop more comprehensive real-world applications with enhanced user interfaces, expanded distress classification capabilities, and integration with existing infrastructure management systems.

Remarks on code availability:

I highly recommend the authors provide some usage examples with jupyter notebooks and real smartphone(android or ios) apps to demonstrate it actually works on those devices.

Response: We appreciate the reviewer's valuable suggestion regarding practical demonstration of the system on real devices.

To support practical reproducibility and verify that the system performs effectively on real-world devices, the complete source code has been made publicly accessible via a dedicated GitHub repository, which includes all necessary components for model deployment.

Additionally, the supplementary materials include two demonstration videos: one showcasing real-time road distress detection on a smartphone, and the other illustrating usage on a Mixed Reality (MR) device. Both videos confirm that the model runs efficiently and accurately on mobile platforms. The accompanying Supplementary Information PDF provides detailed documentation of the deployment pipeline, including model conversion to TFLite format, integration into mobile applications, and field validation results. These materials together substantiate the claim that the proposed framework is fully functional and capable of supporting real-time operation on both Android smartphones and MR systems.

Reviewer #3:

Remarks to the Author:

Based on the revised version and the authors' responses to the reviewers' comments, the author has addressed all the questions and has particularly improved the structure of the article. In this work, the authors proposed a new version of MobileNetV2 that is optimized for real-time detection of road distress. While there is still room for further improvement, considering all the enhancements made in this updated version, I accept the paper for publication.

Response: We sincerely thank the reviewer for the positive evaluation and recognition of the revisions and structural improvements.

Remarks on code availability:

The codes are reorganized and commented to improve their comprehension and be reused by the scientific community.

Response: We are especially grateful for the acknowledgment of the model optimization and deployment contributions. We are encouraged by your support and will continue to refine the framework in future work.

Paper Review

Advancing Infrastructure Monitoring: Lightweight Deep Learning for Real-Time Road Distress Detection on Mobile Devices

Paper summary:

The authors propose a solution for infrastructure monitoring by developing an automated defect detection system based on lightweight deep learning. The system combines an optimized MobileNet V2 model, which has been reduced by 65% while maintaining 92% accuracy to enable real-time detection of road degradation. Tested in Aachen, Germany, the system demonstrated 86% accuracy in real-world conditions, with a fast processing time of 50.11 ms per image and 60% reduced energy consumption compared to the original model. This solution, which integrates smartphones and mixed reality devices with a cloud database, represents an advancement in road infrastructure maintenance, offering a more efficient approach to road monitoring and maintenance.

Comments for the authors:

The article is very interesting as it presents an advancement in the real-time domain using object detection deep learning techniques. However, the paper is poorly organized and should be divided into different sections to simplify the lecturer's work. For this reason I accept the paper for publication after major corrections.

Additional consideration:

The authors should pay attention to:

- To maintain real-time application, why authors don't use quantization techniques. It's important to discuss this point in the paper.
- The authors are invited to add a paragraph that introduce the different sections of the paper.
- After the introduction, the authors should explain their contribution on the proposed models Mobilenet2 and YOLOv8n in a new section.
- It's difficult to read this paper as presented. The whole paper and sections of the paper should be reorganized and devised.